# DiFlow-TTS: Compact and Low-Latency Zero-Shot Text-to-Speech with Factorized Discrete Flow Matching

## Abstract

Despite flow matching and diffusion models having emerged as powerful generative paradigms that advance zero-shot text-to-speech (TTS) systems in continuous settings, they continue to fall short in capturing high-quality speech attributes such as naturalness, similarity, and prosody. A key reason for this limitation is that continuous representations often entangle these attributes, making fine-grained control and generation more difficult. Discrete codec representations offer a promising alternative, yet most flow-based methods embed tokens into a continuous space before applying flow matching, diminishing the benefits of discrete data. In this work, we present DiFlow-TTS, which, to the best of our knowledge, is the first model to investigate discrete flow matching directly to generate high-quality speech from discrete inputs. Leveraging factorized speech attributes, DiFlow-TTS introduces a factorized flow prediction mechanism that simultaneously predicts prosody and acoustic detail through separate heads, enabling explicit modeling of aspect-specific distributions. Experimental results demonstrate that DiFlow-TTS delivers strong performance across several metrics, while maintaining a compact model size up to 11.7 times smaller and low-latency inference that generates speech up to 34 times faster than recent state-of-the-art baselines. Code and audio samples are available on our demo page: `https://diflow-tts.github.io`.

## 1 Introduction

Zero-shot TTS has made remarkable progress in recent years, with the goal of generating high-quality speech that faithfully replicates the voice of previously unseen speakers from only a few seconds of reference audio. Recent studies have explored autoregressive approaches, where speech is quantized into discrete tokens and modeled using language models (Zhang et al., 2023; Han et al., 2024; Meng et al., 2025; Song et al., 2024; Chen et al., 2024a; Peng et al., 2024; Ji et al., 2024a; Wang et al., 2025b; Chen et al., 2025). While these models achieve strong performance in terms of naturalness and speaker similarity, they generally require large-scale training data to be effective. Furthermore, their autoregressive nature leads to slow inference and introduces common decoding error patterns, such as unintended repetitions of reference content or omissions of initial words in the input text.

To overcome these limitations, non-autoregressive (NAR) approaches have been developed, enabling faster generation through parallel decoding. Among these, diffusion-based (Kang et al., 2023; Shen et al., 2024; Ju et al., 2024; Lee et al., 2025) and flow-based (Kim et al., 2023; Le et al., 2023; Mehta et al., 2024; Eskimez et al., 2024; Chen et al., 2024b) models have emerged as effective generative frameworks for TTS, striking a better balance between synthesis quality and inference efficiency. These models typically operate in the mel-spectrogram domain, which preserves rich acoustic detail and enables in-context learning via target speech prompting, leading to improved speaker similarity. However, zero-shot TTS requires more than voice cloning - it demands precise modeling of multiple speech attributes. A natural solution is to factorize the reference speech into attributes such as prosody, content, and acoustic details, and to model each of these components explicitly. In this manner, continuous representation often entangle these attributes, hindering fine-grained manipulation.

Recent efforts to adapt discrete codec tokens to generative paradigms have sparked a growing interest in applying diffusion models within fully discrete settings (Yang et al., 2023; Wu et al., 2024; Yang et al., 2024; Ju et al., 2024). In contrast, flow matching models designed for discrete data typically

follow a single approach: first embedding the discrete inputs into a continuous space, then applying continuous flow matching within that space (Du et al., 2024b; Hieu et al., 2025; Wang et al., 2025a; Zuo et al., 2025a). This approach introduces redundant operations in the transition between discrete and continuous representations, complicating the training process. A direct discrete formulation removes this conversion, reduces compute and memory demands, and avoids instability arising from arbitrary embedding choices. Although Discrete Flow Matching (DFM) has shown potential in language, vision, and bioinformatics (Gat et al., 2024; Shaul et al., 2025; Yadav et al., 2025; Fuest et al., 2025), its application to speech synthesis remains unexplored. This motivates our key question:

> Can we harness purely discrete flow matching on factorized codec tokens to achieve high-fidelity speech synthesis?

In this study, to investigate the viability of Factorized **Di**screte **Flow** Matching in zero-shot **TTS**, we present DiFlow-TTS, as illustrated in Figure 1, to directly address the aforementioned question. The core of our approach lies in modeling and controlling the distributions of discrete factorized speech representations, providing a principled framework for synthesizing high-quality speech in discrete settings. To this end, we leverage FACodec (Ju et al., 2024) to decompose speech into prosody, content, and acoustic tokens, serving as our target discrete data. Building on this, DiFlow-TTS explicitly models these factorized attributes within a compact and unified framework. Specifically, we design Phoneme-Content Mapper (PCM), which maps phoneme sequences to discrete speech tokens that represent the content of the utterance.

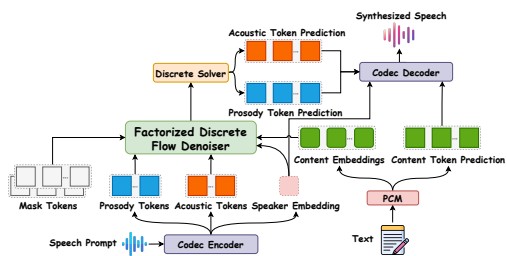

Figure 1: Overview of DiFlow-TTS. A Codec Encoder decomposes the speech prompt into speaker, prosody, and acoustic tokens, while the *Phoneme-Content Mapper* converts text into content embeddings. Conditioned on these, the *Factorized Discrete Flow Denoiser* generates prosody and acoustic tokens, and the Codec Decoder reconstructs the waveform.

This module generates content embeddings that align closely with the semantic structure of the speech. These embeddings, along with auditory attributes extracted from the reference speech prompt, are then used to condition a Factorized Discrete Flow Denoiser (FDFD) module, allowing it to effectively clone the reference's speaking style. Crucially, we design the model with separate prediction heads for the probability velocity of distinct speech aspects, specifically prosody, and acoustic details, allowing it to learn aspect-specific distributions explicitly. As a result, DiFlow-TTS delivers enhanced naturalness, expressiveness, and speaker fidelity. Our main contributions are as follows:

- We introduce DiFlow-TTS, the first zero-shot TTS framework that learns probability flows directly in the discrete space of factorized codec tokens. This removes the continuous embedding detour used by prior "discrete" flow methods, accelerating inference process.

- We present Phoneme-Content Mapper, which aligns phoneme sequences to discrete content tokens, providing precise semantic grounding that guides the generation of prosody and acoustic attributes.

- We propose a Factorized Discrete Flow Denoiser that explicitly models individual speech attributes through the flow-prediction mechanism with dedicated heads for prosody and acoustic details, enabling explicit learning of aspect-specific distributions within a compact and unified architecture, without the need for multiple generators.

- We demonstrate that DiFlow-TTS outperforms baselines in naturalness, content accuracy, and prosody preservation while maintaining a compact model size, up to 11.7 times smaller, and achieving low-latency inference, up to 34 times faster than baselines, making it suitable for resource-constrained, latency-sensitive systems.

## 2 RELATED WORK

A growing trend in speech synthesis focuses on converting raw waveforms into discrete token representations using vector-quantized variational autoencoders (VQ-VAE), which was first introduced by

(van den Oord et al., 2017) in the field of computer vision and later adapted to speech processing (Baevski et al., 2020; Hsu et al., 2021). These tokenized representations have demonstrated greater naturalness and robustness compared to conventional mel-spectrogram-based approaches. To effectively model sequences of discrete speech tokens, recent efforts have adapted large language models (LLMs) from the natural language processing (NLP) domain (Zhang et al., 2023; Chen et al., 2024a; Han et al., 2024; Du et al., 2024b; Peng et al., 2024; Meng et al., 2025; Chen et al., 2025; Wang et al., 2025b). A notable example is VALL-E (Chen et al., 2025), which leverages a pre-trained neural codec to encode speech into discrete codec tokens and reformulates zero-shot TTS as a conditional codec language modeling task. During inference, it performs autoregressive continuation from the acoustic tokens of a short speech prompt, enabling high-fidelity speaker-consistent voice synthesis.

Although autoregressive models achieve impressive quality, they are inherently limited by slow inference speeds. This limitation has prompted a shift toward NAR paradigms (Shen et al., 2024; Ju et al., 2024; Du et al., 2024a; Lee et al., 2025; Jia et al., 2025). For example, NaturalSpeech 2 (Shen et al., 2024) uses diffusion (Ho et al., 2020; Song et al., 2021) to generate discrete acoustic tokens as continuous features. Its successor, NaturalSpeech 3 (Ju et al., 2024), further factorizes speech into subspaces of content, prosody, and acoustic details, employing multiple diffusion models to independently capture various acoustic characteristics. In parallel, flow matching (Lipman et al., 2023; Liu et al., 2023) has gained attention as a promising generative technique, producing strong results in various domains. However, most existing speech-related flow matching applications operate in a continuous space (Mehta et al., 2024; Guan et al., 2024; Yao et al., 2025; Zuo et al., 2025b;a; Hieu et al., 2025), requiring either a pure mel-spectrogram or discrete tokens to be embedded into continuous representations prior to generation. An emerging line of research seeks to extend iterative refinement techniques to discrete spaces by modeling generation dynamics using Markov chains. Discrete-space generative models have already proven effective in domains such as natural language (Lou et al., 2024; Shi et al., 2024; Sahoo et al., 2024), proteins (Campbell et al., 2024; Yi et al., 2025), vision (Austin et al., 2021; Chang et al., 2022; Shi et al., 2024; Fuest et al., 2025), code (Gat et al., 2024), and even graphs (Qin et al., 2025). Although discrete diffusion models have recently been applied to speech synthesis (Ye et al., 2025; Ye & Shan, 2025), the use of discrete flow matching (Gat et al., 2024) to model speech tokens remains largely unexplored, particularly in zero-shot TTS scenarios. In this work, we propose a DFM framework tailored for zero-shot TTS, aiming to harness the efficiency of discrete modeling without compromising quality.

# 3 METHODOLOGY

Figure 2 illustrates the overall framework of DiFlow-TTS, which comprises three main modules: (a) *Speech Tokenization*, (b) *Phoneme-Content Mapper*, and (c) *Factorized Discrete Flow Denoiser*. In the following sections, we describe each module in detail.

## 3.1 PRELIMINARIES

**Notation.** Let a sequence $x$ be an array of $L$ tokens $(x^1, x^2, \ldots, x^L)$ drawn from a discrete vocabulary of size $v$, i.e., $x \in \mathcal{D} = [v]^L$ with $[v] = \{1, \ldots, v\}$. We further define the extended space $\mathcal{D}' = [v]^{nL}$ as the concatenation of $n$ such sequences. To represent the point mass distributions over these sequences, we use the delta function $\delta_y(x) = \prod_{i=1}^{L} \delta_{y^i}(x^i)$, where $y \in \mathcal{D}$, $\delta_{y^i}(x^i) = 1$ if $x^i = y^i$, and 0 otherwise.

**Discrete Flow Matching.** We adopt DFM as the generative backbone for codec token modeling. The goal is to transport source samples $\mathbf{x}_0 \sim p$ to target samples $\mathbf{x}_1 \sim q$. Concretely, we instantiate the source distribution with all-mask tokens, while the target distribution is factorized into prosodic and acoustic components, enabling structured joint learning. During training, we employ a scheduler $\kappa_t \in [0, 1]$, a monotonically increasing function with boundary conditions $\kappa_0 = 0$ and $\kappa_1 = 1$, where $t \in [0, 1]$ denotes continuous time. This scheduler controls the interpolation, gradually shifting the distribution from source to target as $\kappa_t$ increases. Following Gat et al. (2024), we then construct a conditional probability path, referred to as the *mixture path*, which linearly interpolates between the source and target distributions: $p_t(\mathbf{x}^i|\mathbf{x}_0, \mathbf{x}_1) = (1 - \kappa_t)\delta_{\mathbf{x}_0}(\mathbf{x}^i) + \kappa_t\delta_{\mathbf{x}_1}(\mathbf{x}^i)$. This formulation leads to a *conditional probability path*, which is governed by the probability velocity $\mathbf{u}_t$ defined as:

$$\mathbf{u}_t^i(\mathbf{x}^i, \mathbf{x}_t) = \frac{\dot{\kappa}_t}{1 - \kappa_t} \left[ p_{1|t}(\mathbf{x}^i|\mathbf{x}_t, \mathbf{c}; \theta) - \delta_{\mathbf{x}_t}(\mathbf{x}^i) \right], \tag{1}$$

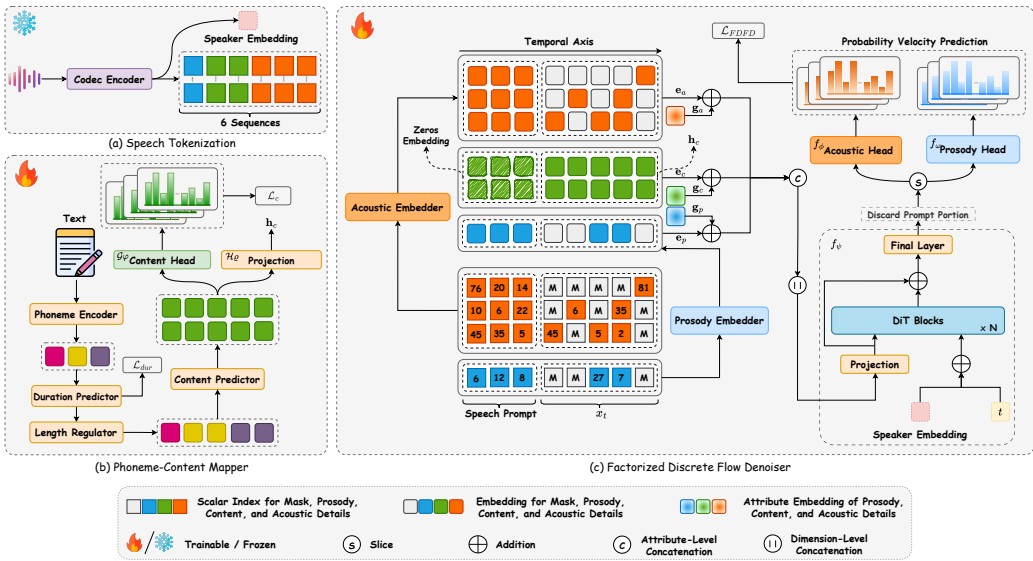

Figure 2: The detailed components of DiFlow-TTS. The architecture consists of three main components: (a) *Speech Tokenization*, which extracts discrete tokens and a speaker embedding from a raw speech; (b) *Phoneme-Content Mapper*, which maps input phonemes to discrete content tokens and generates the corresponding content embeddings; and (c) *Factorized Discrete Flow Denoiser*, which performs discrete flow matching conditioned on the content embeddings, speaker embedding, and the discrete prosody and acoustic tokens derived from the reference speech prompt.

where $\dot{\kappa}_t$ is the time derivative of the scheduler $\kappa_t$, $\theta$ denotes learnable parameters of a probability denoiser, $p_{1|t}(\cdot|\mathbf{x}_t, \mathbf{c}; \theta)$ is the posterior distribution $\mathbf{x}_1$ given a partially corrupted sequence $\mathbf{x}_t$ and $\mathbf{c}$ representing a set of multimodal conditioning inputs. More details are provided in Section B.1.

## 3.2 SPEECH TOKENIZATION

The Speech Tokenization module (Figure 2a) converts a raw input speech waveform into distinct token sequences. For this process, we employ FACodec Ju et al. (2024), which factorizes the original speech signal $\mathbf{r}$ into disentangled token sequences representing prosody, content, and acoustic details and extracts the speaker identity:

$$\mathbf{r}^p, \mathbf{r}^c, \mathbf{r}^a, \mathbf{s} = \text{CodecEncoder}(\mathbf{r}), \tag{2}$$

where $\mathbf{r}^p \in [v]^{mL}$, $\mathbf{r}^c \in [v]^{nL}$, and $\mathbf{r}^a \in [v]^{kL}$ denote the discrete token sequences for prosody, content, and acoustic details, respectively, and $\mathbf{s} \in \mathbb{R}^{D_{\text{spk}}}$ is the embedding of the speaker. Here, $m$, $n$, and $k$ represent the number of sequences representing each attribute, and $L$ is the sequence length. More details are provided in Section B.2.

## 3.3 PHONEME-CONTENT MAPPER

The PCM module (Figure 2b) transforms the phonemes derived from the text prompt into content tokens corresponding to those that would be generated by the speech tokenizer, along with the corresponding content embeddings.

Given a text prompt, we first use a grapheme-to-phoneme converter[1] to obtain the textual phoneme sequence $\mathbf{P} = (\mathbf{P}_1, \mathbf{P}_2, \ldots, \mathbf{P}_N)$ consisting of $N$ tokens. A phoneme encoder then transforms $\mathbf{P}$ into a sequence of embeddings $\mathbf{p} = (\mathbf{p}_1, \mathbf{p}_2, \ldots, \mathbf{p}_N) \in \mathbb{R}^{N \times D}$, where $D$ denotes the hidden dimension. To align phonemes with discrete speech tokens, a Duration Predictor estimates the duration $\mathbf{d} = (\mathbf{d}_1, \mathbf{d}_2, \ldots, \mathbf{d}_N)$, indicating how many speech tokens correspond to each phoneme. This produces an integer-based alignment that maps each phoneme to a variable-length span in the speech-token sequence. Using these alignments, the Length Regulator upsamples phoneme

---

[1] https://github.com/Kyubyong/g2p

embeddings to match the length of the discrete content token sequence $L$. The upsampled sequence is then passed to the Content Predictor (Figure 6), which consists of multiple Feed-Forward Transformer (FFT) layers. These layers hierarchically extract $n$ content representations, producing hidden states $\mathbf{h} \in \mathbb{R}^{n \times L \times D}$, which are then processed by two branches: a projection layer $\mathcal{H}_\varrho(\cdot)$ produces content embeddings, and a content head $\mathcal{G}_\varphi(\cdot)$ that outputs logits over a vocabulary of size $v$:

$$\mathbf{h}_c = \mathcal{H}_\varrho(\mathbf{h}) \in \mathbb{R}^{n \times L \times D},$$
$$p(\mathbf{x}^c|\mathbf{h}; \varphi) = \mathcal{G}_\varphi(\mathbf{h}) \in \mathbb{R}^{n \times L \times v}. \tag{3}$$

## 3.4 Factorized Discrete Flow Denoiser

The FDFD (Figure 2c) aims to generate the prosody and acoustic sequences of the target speech by leveraging DFM and in-context learning, conditioned on a set of contextual inputs. In the following, we detail the key elements of this module.

**Contextual Modeling.** We now elaborate on the construction of the conditioning context $\mathbf{c}$ introduced in Equation (1) and describe how it is integrated into our framework.

Given a reference speech prompt $\mathbf{r}$, we decompose it as shown in Equation (2) into a prosody token sequence $\mathbf{r}^p \in [v]^{mL_p}$, an acoustic token sequence $\mathbf{r}^a \in [v]^{kL_p}$, and a speaker embedding $\mathbf{s} \in \mathbb{R}^{D_{\mathrm{spk}}}$, where $L_p$ denotes the temporal length of the reference prompt, and $D_{\mathrm{spk}}$ is the hidden dimension of the speaker embedding. Likewise, the current denoising input $\mathbf{x}_t \in [v]^{(m+k)L}$ is split into prosody tokens $\mathbf{x}_t^p \in [v]^{mL}$ and acoustic tokens $\mathbf{x}_t^a \in [v]^{kL}$. We then use prosody and acoustic embedders, denoted $\mathcal{E}_p(\cdot)$ and $\mathcal{E}_a(\cdot)$, to convert these sequences into hidden representations:

$$\mathbf{e}_r^p = \mathcal{E}_p(\mathbf{r}^p) \in \mathbb{R}^{m \times L_p \times D}, \quad \mathbf{e}_t^p = \mathcal{E}_p(\mathbf{x}_t^p) \in \mathbb{R}^{m \times L \times D},$$
$$\mathbf{e}_r^a = \mathcal{E}_a(\mathbf{r}^a) \in \mathbb{R}^{k \times L_p \times D}, \quad \mathbf{e}_t^a = \mathcal{E}_a(\mathbf{x}_t^a) \in \mathbb{R}^{k \times L \times D}.$$

To enrich the modeling of prosody and acoustic information, we further incorporate the embedding of content $\mathbf{h}_c$ obtained from Equation (3) and the embedding of speakers $\mathbf{s}$ extracted from the reference speech prompt described above. Specifically, for each attribute, the reference embedding $\mathbf{e}_r^i$ is concatenated with its corresponding corrupted embedding $\mathbf{e}_t^i$, where $i \in \{p, c, a\}$ denotes prosody, content, and acoustic details, respectively. For the corrupted content embedding $\mathbf{e}_t^c$, we directly use the content representation: $\mathbf{e}_t^c = \mathbf{h}_c \in \mathbb{R}^{n \times L \times D}$. Since content information is not required for the reference branch, we set $\mathbf{e}_r^c$ to a zero-valued placeholder $\mathbf{h}_{\mathrm{zeros}} \in \mathbb{R}^{n \times L_p \times D}$ to maintain consistency in the number of quantizer streams. The concatenated embeddings for each attribute are given by:

$$\mathbf{e}_p = \mathbf{e}_r^p \oplus \mathbf{e}_t^p \in \mathbb{R}^{m \times (L_p+L) \times D},$$
$$\mathbf{e}_c = \mathbf{e}_r^c \oplus \mathbf{e}_t^c \in \mathbb{R}^{n \times (L_p+L) \times D},$$
$$\mathbf{e}_a = \mathbf{e}_r^a \oplus \mathbf{e}_t^a \in \mathbb{R}^{k \times (L_p+L) \times D},$$

where $\oplus$ denotes the concatenation of the sequence.

To help the model distinguish among attribute types, we introduce learnable attribute-type embeddings: $\mathbf{g}_p$, $\mathbf{g}_c$, and $\mathbf{g}_a$, each in $\mathbb{R}^{1 \times 1 \times D}$, corresponding to prosody, content, and acoustic details attributes, respectively. These are broadcast and added to their respective embeddings to inject attribute-type awareness. The resulting embeddings are then concatenated along the temporal dimension as follows:
$$\mathbf{e} = [(\mathbf{e}_p + \mathbf{g}_p) \oplus (\mathbf{e}_c + \mathbf{g}_c) \oplus (\mathbf{e}_a + \mathbf{g}_a)] \in \mathbb{R}^{(m+n+k) \times (L_p+L) \times D}.$$

We reshape the combined embedding $\mathbf{e}$ by permuting its axes to flatten the dimension of the quantizer, resulting in a tensor of shape $\mathbb{R}^{(L_p+L) \times (m+n+k)D}$. This reshaped embedding is then projected into the model's hidden dimension $D$, yielding $\mathbf{z} \in \mathbb{R}^{(L_p+L) \times D}$, through a learnable projection layer. The resulting sequence $\mathbf{z}$ is passed through a neural network $f_\psi : \mathbb{R}^{(L_p+L) \times D} \to \mathbb{R}^{(L_p+L) \times (m+k)D}$, implemented with Diffusion Transformer (DiT) blocks Peebles & Xie (2023). In parallel, the timestep $\mathbf{t}$ is embedded into $\mathbb{R}^D$ and added to the speaker embedding $\mathbf{s}$, which is also projected into $\mathbb{R}^D$, to form a global conditioning vector. This vector is fed into a multilayer perceptron (MLP), which outputs scaling and shifting parameters used for feature-wise affine modulation, enabling speaker-aware adaptation. After residual addition, the final transformation is applied, comprising layer normalization followed by feature-wise affine modulation conditioned on the global conditioning vector, and a linear projection to $(m+k)D$. We then discard the reference portion and permute the result to yield the final hidden representation $\mathbf{h}_{p,a} \in \mathbb{R}^{(m+k) \times L \times D}$.

**Factorized Flow Prediction.** To effectively enables the model to jointly attend to information from different representation subspaces, we propose a factorized flow prediction mechanism based on multi-head prediction. In this design, FDFD simultaneously models multiple aspects of speech, specifically prosody and acoustic details. Formally, we define two parallel heads: the *prosody head* $f_\phi(\cdot)$ and the *acoustic head* $f_\omega(\cdot)$, which independently predict probability distributions corresponding to prosody and acoustic attributes. We begin by slicing the representation $\mathbf{h}_{p,a}$ into two parts: the prosody representation $\mathbf{h}_p \in \mathbb{R}^{m \times L \times D}$ and the acoustic representation $\mathbf{h}_a \in \mathbb{R}^{k \times L \times D}$. Each component is processed by its respective head, $f_\phi(\cdot)$ and $f_\omega(\cdot)$, producing logits of the shapes $\mathbb{R}^{m \times L \times v}$ and $\mathbb{R}^{k \times L \times v}$, respectively. These logits correspond to the categorical distributions predicted over the discrete token vocabulary for each aspect. Finally, the two outputs are concatenated along the dimension of the quantizer, producing a unified tensor of shape $\mathbb{R}^{(m+k) \times L \times v}$. This tensor serves as the estimated posterior distribution over $\mathbf{x}_1$.

### 3.5 TRAINING OBJECTIVES

Our training objective integrates three loss components, one for each module in the framework. First, we optimize the *Duration Predictor* using a loss of the Mean Squared Error (MSE) on a logarithmic scale, denoted as $\mathcal{L}_{dur}$, which compares the predicted and ground-truth durations. Second, for the *Content Predictor* defined in Equation (3), we use a cross-entropy loss $\mathcal{L}_c$ between the predicted logits and the discrete content tokens obtained from the ground truth. Third, for the *FDFD* module, we learn a probabilistic denoiser $p_{1|t}$ trained to recover masked tokens under varying masking ratios. The objective is to minimize the cross-entropy loss: $\mathcal{L}_{FDFD}(\theta) = -\sum_{i \in \mathcal{T}} \mathbb{E}_{t \sim \mathcal{U}[0,1], (\mathbf{x}_0, \mathbf{x}_1), \mathbf{x}_t} \left[ \log p_{1|t}(\mathbf{x}_1^i | \mathbf{x}_t, \mathbf{c}; \theta) \right]$, where $\mathcal{T} = [(m+k)L]$, $\mathbf{x}_t \sim p_t(\mathbf{x}|\mathbf{x}_0, \mathbf{x}_1), \mathbf{x}_0 \sim p$, and $\mathbf{x}_1 \sim q$. Finally, the total loss is defined as:

$$\mathcal{L} = \lambda_{dur}\mathcal{L}_{dur} + \lambda_c\mathcal{L}_c + \lambda_{FDFD}\mathcal{L}_{FDFD}, \tag{4}$$

where $\lambda_{dur}$, $\lambda_c$, and $\lambda_{FDFD}$ are hyperparameters weighting the loss terms. Algorithms 1 and 2 further describe our training and inference pipeline.

## 4 EXPERIMENTS

### 4.1 EXPERIMENTAL SETUP

**Implementation Details.** The FDFD uses a scheduler $\kappa_t$ drawn from a family of cubic polynomials. In our implementation, we set $\kappa_t = 1 - t^2$. The module employs DiT blocks (Peebles & Xie, 2023) with a hidden size of 768, 12 layers, and 12 attention heads, further enhanced with rotary position embedding (RoPE) (Su et al., 2024). We train the model on $4\times$ NVIDIA A100 GPUs for 315K steps with a batch size of 16, using AdamW (Loshchilov & Hutter, 2019) with a learning rate of $1 \times 10^{-4}$, weight decay of 0.01, and 200K warm-up steps. The overall objective combines duration, content, and denoising losses, weighted by $\lambda_{dur} = 0.5$, $\lambda_c = 1.0$, and $\lambda_{FDFD} = 1.0$, respectively. More details are provided in Section A.

**Baselines.** To ensure fairness in evaluation, we compare against publicly available baselines spanning different modeling paradigms: **(i)** *Autoregressive models:* VoiceCraft (Peng et al., 2024), VALL-E (Chen et al., 2025); **(ii)** *Continuous flow matching/diffusion models:* NaturalSpeech 2 (Shen et al., 2024), F5-TTS (Chen et al., 2024b), OZSpeech (Hieu et al., 2025); **(iii)** *Masked generative model:* MaskGCT (Wang et al., 2025c). We benchmark these systems against our proposed method, **(iv)** *Discrete flow matching model:* DiFlow-TTS. More details are provided in Section C.3.

**Dataset.** We use a 470-hour subset of the *LibriTTS* dataset (Zen et al., 2019), which consists of multi-speaker English audio recordings, to train our method. For a fair comparison, we reproduce the baselines F5-TTS and VALL-E using a 500-hour subset of *LibriTTS*. Due to the complexity of reproduction, we directly use the released checkpoints of NaturalSpeech 2, trained on 585 hours of *LibriTTS*, as well as VoiceCraft and MaskGCT, which were trained on much larger corpora of approximately 9K hours and 100K hours, respectively. This setup highlights the effectiveness of our approach under limited data conditions, while also allowing comparison against baselines trained on a similar data scale. Additional details are provided in Section C.1.

| Type | Model | Data (hours) | UTMOS ↑ | WER ↓ | SIM-O ↑ | F0 | | Energy | |
|------|-------|--------------|---------|-------|---------|----|----|--------|----|
| | | | | | | Accuracy ↑ | RMSE ↓ | Accuracy ↑ | RMSE ↓ |
| - | Ground Truth | - | 4.10 | 0.02 | - | - | - | - | - |
| (i) | VoiceCraft [†] | GS (9K) | 3.55 | 0.18 | 0.51 | 0.78 | 17.22 | 0.44 | 0.010 |
| | VALL-E [◇] | LT (500) | 3.68 | 0.19 | 0.40 | 0.75 | 21.66 | 0.36 | 0.020 |
| (ii) | NaturalSpeech 2 [‡] | LT (585) | 2.38 | 0.09 | 0.31 | 0.80 | 15.62 | 0.25 | 0.020 |
| | F5-TTS [◇] | LT (500) | 3.76 | 0.24 | 0.52 | 0.80 | 13.78 | 0.67 | 0.010 |
| | OZSpeech [†] | LT (500) | 3.15 | **0.05** | 0.40 | 0.81 | 11.96 | 0.67 | 0.010 |
| (iii) | MaskGCT [†] | E (100K) | 3.83 | 0.09 | **0.67** | 0.77 | 14.33 | **0.75** | **0.007** |
| (iv) | **DiFlow-TTS** | **LT (470)** | **3.98** | **0.05** | 0.45 | **0.88** | **7.97** | 0.73 | **0.007** |

Table 1: Performance on the *LibriSpeech test-clean* dataset using 3-second audio prompts. [◇] means reproduced results. [†] and [‡] mean results inferred from official and unofficial checkpoints, respectively. The best and second best are **bold** and underlined, respectively. Abbreviation: E (Emilia), GS (GigaSpeech), LT (LibriTTS).

**Evaluation Metrics.** To evaluate model performance, we use a range of *objective evaluation* metrics targeting various aspects: naturalness and speech quality is measured with UTMOS; speaker similarity is evaluated using SIM-O; robustness is reflected by the word error rate (WER); and prosody accuracy and error are analyzed through pitch and energy metrics. In addition, we assess model latency using the real-time factor (RTF). More details on these metrics are provided in Section C.2. Along with these objective evaluations, we perform a *subjective assessment* based on the Mean Opinion Score (MOS) protocol. In this evaluation, 30 listeners rate synthesized speech on a scale from 1 to 5 based on naturalness, intelligibility, and speaker similarity to the speech prompt.

## 4.2 Main Results

**Comparison Results.** Table 1 presents the performance of DiFlow-TTS with 128 function evaluations (NFE) using 3-second audio prompts, compared to baseline methods. DiFlow-TTS achieves superior naturalness and speech quality, as measured by UTMOS, despite being trained on only 470 hours of speech data, which is significantly less ($1.1\times$ to $212.8\times$) than other baselines, highlighting the strength of our FDFD module in capturing prosodic and acoustic nuances even under limited data conditions. In terms of linguistic accuracy, DiFlow-TTS, along with OZSpeech, achieves SOTA perfor-

| Model | Naturalness ↑ | Intelligibility ↑ | Similarity ↑ |
|-------|---------------|-------------------|--------------|
| Ground Truth | $4.42 \pm 0.12$ | $4.54 \pm 0.11$ | $4.29 \pm 0.14$ |
| VoiceCraft | $3.94 \pm 0.17$ | $4.08 \pm 0.18$ | $4.17 \pm 0.15$ |
| VALLE-E | $3.71 \pm 0.17$ | $3.96 \pm 0.17$ | $3.99 \pm 0.15$ |
| NaturalSpeech 2 | $2.62 \pm 0.20$ | $3.25 \pm 0.21$ | $2.63 \pm 0.18$ |
| F5-TTS | $3.97 \pm 0.17$ | $4.16 \pm 0.14$ | $4.07 \pm 0.16$ |
| OZSpeech | $2.80 \pm 0.23$ | $3.42 \pm 0.24$ | $3.20 \pm 0.22$ |
| MaskGCT | $3.97 \pm 0.16$ | $4.14 \pm 0.15$ | $4.17 \pm 0.15$ |
| **DiFlow-TTS** | **$4.18 \pm 0.16$** | **$4.41 \pm 0.13$** | **$4.42 \pm 0.12$** |

Table 2: MOS evaluation with 3-second audio prompts, including 95% confidence intervals. The best and second best are **bolded** and underlined, respectively.

mance in terms of WER, demonstrating the effectiveness of our method in producing speech with accurate linguistic content. For speaker similarity, DiFlow-TTS offers no clear advantage over baselines, likely due to its simple speaker conditioning in DiT blocks, which could be improved with more advanced strategies. For prosody reconstruction, DiFlow-TTS outperforms across all metrics, with the sole exception of energy accuracy, where it trails MaskGCT by only 0.02 despite MaskGCT being substantially larger and trained on significantly more data. These findings further confirm the ability of the FDFD module to model fine-grained prosodic attributes with high fidelity. To gain further insight into speech quality, we report the subjective MOS evaluations in Table 2. These results are consistent with the findings in Table 1. Overall, DiFlow-TTS consistently outperforms SOTA methods across every MOS dimensions, providing strong evidence of its well-balanced performance in generating natural and intelligible speech with high speaker similarity. It is worth noting that even though DiFlow-TTS ranks third on SIM-O, an embedding-space proxy that may penalize artifacts inaudible to humans, it best captures the perceptual identity cues (e.g., pitch, timbre, prosody) that listeners value, indicating superior speaker faithfulness where it matters most. These results are especially notable given the model's training data efficiency.

**Model Size & Latency Analysis.** Table 3 compares the model size and latency between DiFlow-TTS and the baselines for the 3-second audio prompt setting. The RTF metric, measured in seconds,

| Model | #Params ↓ | NFE | RTF ↓ | UTMOS ↑ | WER ↓ | SIM-O ↑ | RMSE$_{F0}$ ↓ | RMSE$_E$ ↓ |
|---|---|---|---|---|---|---|---|---|
| VoiceCraft | 830M | - | 1.70 | 3.55 | 0.18 | 0.51 | 17.22 | 0.010 |
| VALL-E | 594M | - | 0.86 | 3.68 | 0.19 | 0.40 | 21.66 | 0.020 |
| NaturalSpeech 2 | 378M | 200 | 1.66 | 2.38 | 0.09 | 0.31 | 15.62 | 0.020 |
| F5-TTS | 336M | 32 | 0.26 | 3.76 | 0.24 | 0.52 | 13.78 | 0.010 |
| OZSpeech | 145M | 1 | **0.03** | 3.15 | **0.05** | 0.40 | 11.96 | 0.010 |
| MaskGCT | 1.43B | 50 + 45$^†$ | 0.46 | 3.83 | 0.09 | **0.67** | 14.33 | **0.007** |
| **DiFlow-TTS-Small** | **122M** | 4 | **0.03** | 3.34 | 0.06 | 0.43 | 8.31 | **0.007** |
| | | 16 | 0.05 | **3.89** | **0.05** | 0.45 | 8.58 | 0.008 |
| **DiFlow-TTS** | 164M | 4 | **0.03** | 3.31 | **0.05** | 0.44 | 8.05 | **0.007** |
| | | 16 | 0.07 | 3.86 | **0.05** | 0.45 | **7.96** | **0.007** |

† MaskGCT is a two-stage system that first predicts masked semantic tokens, then uses them to infer masked acoustic tokens.

Table 3: Comparison of model size and latency. The **#Params** exclude the neural codec or vocoder component, which is non-trainable. The best and second best are **bold** and underlined, respectively.

shows that all baselines except OZSpeech experiences latency in the order of hundreds of miliseconds. In contrast, DiFlowTTS, along with OZSpeech, has latency that is an order of magnitude smaller. To highlight efficiency, we further construct a smaller variant of DiFlow-TTS (denoted as *DiFlow-TTS-Small*) by reducing the number of attention heads ($12 \rightarrow 8$) and DiT layers ($12 \rightarrow 8$), resulting in a 122M-parameter model. This small variant achieves the best results in both speed and size. For comparison, OZSpeech, optimized for the 1-NFE setting, achieves the same RTF as our small model with 4 NFEs, yet delivers significantly lower performance across all metrics except for WER where DiFlow-TTS achieves comparable performance. Furthermore, with 16 NFEs, DiFlow-TTS-Small achieves competitive performance in naturalness, intelligibility, speaker similarity, and prosody error while being only marginally slower than OZSpeech (by 0.02s in RTF) yet 5.2× to 34.0× faster than the other baselines, with a model size 1.2× to 11.7× smaller than all baselines. The results from DiFlow-TTS further reinforces these findings, demonstrating a strong balance between model size, speed, and speech quality.

### 4.3 Ablation Studies and Analyses

**Effect of NFE.** We investigate the impact of varying NFE from 1 to 128 on DiFlow-TTS performance to explore the trade-off between inference efficiency and synthesis quality, as presented in Table 4. Increasing NFE markedly improves UTMOS, indicating that the FDFD module benefits from additional refinement steps to generate more natural speech. In particular, performance stabilizes around 32 NFE, with optimal audio quality observed at 64 NFE, and only marginal improvements beyond this point. Although RTF naturally increases with NFE, the overall latency remains competitive (see Table 3). These results demonstrate its effective trade-off between quality and efficiency.

| NFE | RTF ↓ | UTMOS ↑ |
|---|---|---|
| 1 | 0.022 | 2.904 |
| 2 | 0.025 | 2.908 |
| 4 | 0.031 | 3.313 |
| 8 | 0.043 | 3.698 |
| 16 | 0.066 | 3.864 |
| 32 | 0.112 | 3.923 |
| 64 | 0.207 | 3.958 |
| 128 | 0.394 | 3.978 |

Table 4: Performance of DiFlow-TTS vs. NFE count with 3-second audio prompts.

**Effect of Each Component.** To assess the impact of each component in DiFlow-TTS, we perform an ablation study by systematically removing or modifying key elements: (1) removing the attribute-type embeddings used to distinguish prosody, content, and acoustic streams; (2) excluding the speaker embedding from the conditioning process (i.e., not injecting it into the DiT blocks); (3) disabling the use of content embeddings in the FDFD module; and (4) replacing the multi-head prediction architecture with a single-head prediction. As shown in Table 5, we observe a slight degradation in all metrics except UTMOS when the attribute-type embeddings are removed. This suggests that while these embeddings enhance overall fidelity and prosody modeling, they may introduce minor redundancies that subtly affect perceived naturalness. A more pronounced decline in speaker similarity and prosody-related metrics is observed when speaker embedding is excluded from the FDFD module. This highlights that prosody is not only content-dependent but also strongly influenced by speaker identity; without speaker conditioning, the FDFD module produces extraneous prosodic variations, resulting in reduced speaker adaptation and overall synthesis quality. When content embeddings from the PCM branch are removed from FDFD, we observe substantial degradation across metrics related to naturalness and speaker similarity. This demonstrates the critical role of content embeddings in conditioning FDFD to generate appropriate prosody and support speaker

| Model | UTMOS ↑ | WER ↓ | SIM-O ↓ | F0 | | Energy | |
|---|---|---|---|---|---|---|---|
| | | | | Accuracy ↑ | RMSE ↓ | Accuracy ↑ | RMSE ↓ |
| **DiFlow-TTS** | 3.978 | **0.048** | **0.454** | **0.884** | **7.972** | **0.735** | **0.007** |
| - w/o Attribute Embedding | **3.983** | 0.060 | 0.444 | 0.869 | 9.289 | 0.712 | 0.008 |
| - w/o Speaker Embedding | 3.902 | 0.057 | 0.378 | 0.681 | 20.868 | 0.615 | 0.010 |
| - w/o Content Embedding | 3.077 | 0.063 | 0.333 | 0.867 | 8.878 | 0.698 | 0.008 |
| - w/o Multi-head Prediction | 3.939 | 0.057 | 0.442 | 0.876 | 8.474 | 0.726 | **0.007** |

Table 5: Ablation study results showing the effect of removing each component from the DiFlow-TTS, with NFE set to 128. The best and second best are **bold** and underlined, respectively.

adaptation. Lastly, replacing the multi-head prediction mechanism with a single-head alternative leads to minor performance drops across all metrics, indicating that the multi-head design enhances prediction diversity and robustness in prosody and acoustic modeling.

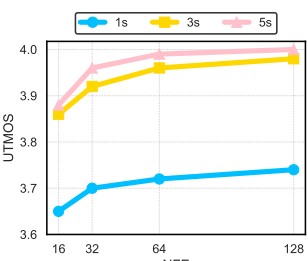

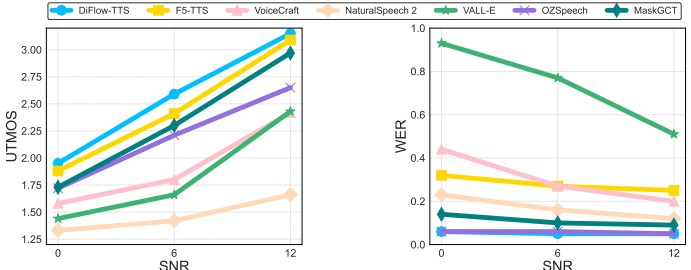

Figure 3: UTMOS vs. NFE for different prompt durations.

Figure 4: Performance across different SNR levels in terms of UTMOS (left) and WER (right).

**Prompt Duration Analysis.** To gain further insight into model behavior, Figure 3 illustrates the relationship between UTMOS, which strongly correlates with human perceptual evaluations, and NFE across different prompt durations. Overall, longer prompts lead to higher UTMOS scores, indicating improved reconstruction quality and a greater sensitivity of the model to prompt length. Additionally, increasing the NFE from 16 to 128 consistently improves performance for all prompt durations. In particular, the highest performance is achieved with a 5-second prompt and an NFE of 128. We refer readers to Section D for detailed comparisons of DiFlow-TTS against baselines under varying prompt lengths.

**Noisy Prompt Analysis.** We evaluate the robustness of DiFlow-TTS under noisy audio prompts using UTMOS and WER, a challenging scenario since most models are trained on clean speech. The noisy prompts are generated from the *LibriSpeech test-clean* set with additive noise augmentation. As shown in Figure 4, all models are highly sensitive to noise, showing sharp degradation in both UTMOS (left) and WER (right) as the prompt SNR decreases (see Table 1 for clean-prompt reference). DiFlow-TTS, however, consistently achieves the highest UTMOS across all noise levels, demonstrating its ability to synthesize high-fidelity speech under noisy conditions. For WER, it shows little to no degradation across SNR levels, a trend also observed in OZSpeech, while other baselines suffer significant performance drops.

## 5 CONCLUSION

In this work, we introduce DiFlow-TTS, a novel zero-shot text-to-speech system that leverages discrete flow matching to model and control the distributions of factorized speech representations, enabling high-quality speech synthesis in discrete settings. By combining a PCM for accurate content modeling with a FDFD that explicitly models prosody and acoustic attributes through aspect-specific heads, DiFlow-TTS delivers strong performance in naturalness, intelligibility, prosody, and synthesis speed, as confirmed by extensive objective and subjective evaluations. These results establish DiFlow-TTS as a compelling solution for efficient, high-fidelity zero-shot speech synthesis, well-suited to resouce-constrained and latency-sensitive applications, and highlight discrete flow models as a promising direction for future generative speech research.

## 6 ETHICS STATEMENT

Our work focuses on advancing text-to-speech (TTS) technology, which, while beneficial, carries potential risks of misuse such as voice spoofing, impersonation, or spreading misleading content. To ensure ethical compliance, all experiments are conducted exclusively on publicly available datasets with appropriate licenses, where speakers have explicitly consented to their voices being used for research. No private or unauthorized data are employed. We acknowledge that the ability to closely mimic a speaker's voice raises important concerns regarding privacy, security, and trust. To mitigate these risks, it is essential to pair progress in TTS with robust detection systems for synthetic speech and to establish mechanisms for reporting and addressing suspected misuse.

## 7 REPRODUCIBILITY STATEMENT

We are committed to reproducible research. Figures 1 and 2 provide both a high-level overview and detailed illustration of the DiFlow-TTS architecture, which is further described in Section 3. To enhance clarity, we also present the algorithmic procedures for training and inference in Algorithms 1 and 2. Comprehensive experimental details are provided across multiple sections: Section C.2 describes the calculation of evaluation metrics, Section C.1 outlines the preprocessing of training and evaluation datasets, and Sections A and 4.1 describe the training and evaluation configurations, ensuring transparency of all parameters. Finally, to facilitate reproducibility and practical use, we release our source code and pretrained weights, accompanied by detailed documentation, via an anonymized GitHub repository linked on the project webpage.

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

## A  IMPLEMENTATION DETAILS

The number of quantizers used in FaCodec (Ju et al., 2024) is $m = 1$ for prosody, $n = 2$ for content, and $k = 3$ for acoustic tokens, each with a vocabulary size of 1024. Figure 5 illustrates the architecture of our DiT block (Peebles & Xie, 2023), which incorporates global conditioning through adaptive normalization. The global conditioning vector, formed by combining time and speaker embeddings, is processed by a Multi-Layer Perceptron (MLP) to generate scale and shift parameters that modulate the input in both the multi-head self-attention (MHSA) and feed-forward stages. The feed-forward network in DiT uses a width multiplier of 4, and the speaker embedding dimension is 256. The *Phoneme-to-Discrete Content Aligner*, shown in Figure 6, has a hidden dimension of 768 and integrates a variance adapter (Ren et al., 2021) with an encoder hidden size of 256, a filter size of 1024, a kernel size of 3, and a dropout rate of 0.5. It employs a hierarchical stack of Feed-Forward Transformer (FFT) blocks, where each level models dependencies conditioned on the outputs of the previous layer. Given phoneme embeddings, the model produces $n$ contextual representations that capture progressively richer features through the stacked FFT layers. Both the text encoder and decoder used to generate content tokens and embeddings adopt the same FFT-based architecture, consisting of 2 layers, 4 attention heads, a hidden size of 256, an output dimension of 768, convolutional filter sizes of 1024 with kernel sizes $[9, 1]$, a dropout rate of 0.2, and a maximum sequence length of 5000.

## B  METHOD DETAILS

### B.1  SOURCE AND TARGET DISTRIBUTIONS

In this section, we elaborate on the source and target distributions of DFT in our setting, as detailed in the following paragraphs.

**Source Distribution** : Following Gat et al. (2024), we instantiate the source distribution $p$ to assign all probability mass to sequences in which every token is the mask token [MASK], that is, $p(x) = \delta_{\text{[MASK]}}(x)$. This implies that the source distribution places all probability mass in the sequence where every token is the mask token [MASK].

**Target Distribution** : In conventional DFM settings, the target sequence $\mathbf{x}_1$ is treated as a monolithic sequence. In contrast, we propose to factorize $\mathbf{x}_1$ into two structured components that are learned jointly. This formulation allows us to construct a probability velocity over a structured target space composed of two parts. To this end, we define the target distribution $q$ as follows:

**Definition B.1.** Let $\mathbf{x}_1^p \sim q_p$ and $\mathbf{x}_1^a \sim q_a$ denote the random variables corresponding to the prosody and acoustic details sequences, respectively. These sequences are in spaces $[v]^{mL}$ and $[v]^{kL}$. The full target sequence is then defined as $\mathbf{x}_1 = \mathbf{x}_1^p \oplus \mathbf{x}_1^a \in [v]^{(m+k)L}$, where $\oplus$ denotes the concatenation of the sequence. Assuming the independence between the two components, the joint target distribution is factorized as $q(x) = q_p(x^p) \cdot q_a(x^a)$, where $x = x^p \oplus x^a$.

## B.2 FACTORIZED NEURAL SPEECH CODEC

The Factorized Neural Speech Codec (FACodec) (Ju et al., 2024) disentangles speech waveforms into distinct attributes, which are content, prosody, acoustic details, and timbre, enabling precise representation for zero shot text-to-speech (TTS) tasks. Given a speech input $x \in \mathbb{R}^C$, a speech encoder, implemented with convolutional blocks, transforms it into a pre-quantization latent representation:

$$h = \text{Encoder}(x) \in \mathbb{R}^{T \times D}, \tag{5}$$

where $T$ represents the downsampled temporal dimension and $D$ denotes the latent feature dimension.

Three factorized vector quantizers (FVQs), denoted $\mathcal{Q}_p$, $\mathcal{Q}_c$, and $\mathcal{Q}_a$ for prosody, content, and acoustic details, respectively, convert $h$ into discrete token sequences. Each FVQ, defined as $\mathcal{Q}_i = \{q_i^j\}_{j=1}^{N_i}$ for $i \in \{p, c, a\}$, consists of $N_i$ quantizers, where $q_i^j \in \mathbb{R}^d$ is the $j$-th quantizer with hidden dimension $d$ and a codebook size of 1024. Specifically, $N_p = 1$, $N_c = 2$, and $N_a = 3$. These quantizers produce discrete codes:

$$z = g_p(h) \oplus g_c(h) \oplus g_a(h) \in \mathbb{R}^{T \times 6}, \tag{6}$$

where $g_p(h) \in \mathbb{R}^{T \times 1}$, $g_c(h) \in \mathbb{R}^{T \times 2}$, and $g_a(h) \in \mathbb{R}^{T \times 3}$ map the latent $h$ to prosody, content, and acoustic detail tokens, respectively. The concatenated output $z$ forms a unified representation of the speech attributes.

The timbre attribute is extracted by passing the hidden representation $h$ through a series of Conformer blocks (Gulati et al., 2020), followed by a temporal pooling layer. This process yields a timbre-specific embedding $z_t \in \mathbb{R}^D$. Given both $z$ and $z_t$, the neural codec decoder reconstructs the speech waveform as follows:

$$y = \text{CodecDecoder}(z, z_t). \tag{7}$$

Building upon the structure of Equation (7), which accepts $z$ and $z_t$ as input and is pre-trained on a large-scale multi-speaker corpus to support robust zero-shot TTS, we propose a method to model and generate a six-dimensional sequence representation $\tilde{z} \in \mathbb{R}^{T \times 6}$. This representation is restricted to lie within the latent subspace of the pre-trained FACodec and is designed to encode prosody, content, and acoustic information in a manner aligned with $z$. Finally, $\tilde{z}$ is passed to $f_{dec}$ along with the timbre embedding $z_t$ to synthesize the output waveform $\tilde{y}$.

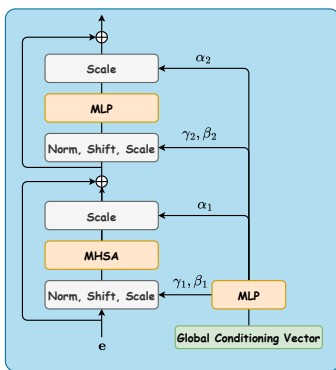

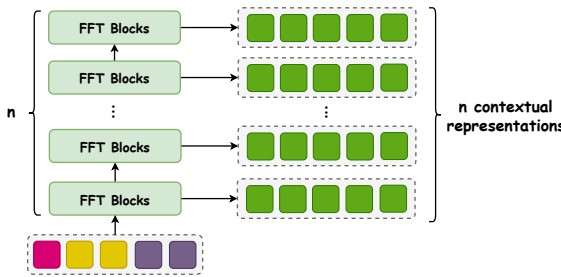

Figure 6: The detailed architecture of Content Predictor.

Figure 5: The detailed architecture of DiT Block.

### B.3 REFERENCE PROMPT SELECTION DURING TRAINING

During training, a crucial step is selecting an appropriate speech prompt to condition the FDFD module. Specifically, we randomly sample an arbitrary segment whose length is 30% of the total temporal length of the ground-truth sequence. This segment serves as the reference prompt, ensuring that prosodic and acoustic characteristics are preserved to guide the FDFD module effectively.

### B.4 TRAINING AND INFERENCE PROCEDURES

To provide a clearer understanding of DiFlow-TTS, we detail the algorithmic procedures for training and inference in Algorithms 1 and 2, respectively.

## C EVALUATION DETAILS

### C.1 DATASET DETAILS

**Training Dataset.** We preprocess the LibriTTS (Zen et al., 2019) dataset for training as follows. The silent segments at the beginning and end of each utterance are removed. We retain audio clips ranging from 1.0 to 16.6 seconds in duration that contain utterances with more than three words. From FACodec, we extract the ground-truth representations, which include a speaker embedding and six sequences of discrete tokens: one for prosody, two for content, and three for acoustic details in order. To obtain the ground truth of phoneme-level text and corresponding discrete speech tokens, we use the Montreal Forced Aligner (MFA) (McAuliffe et al., 2017) to align each audio with its target transcription, producing the duration of each phoneme in the audio. We then multiply these durations by 80, which represents the number of tokens per second in FACodec, to determine the number of speech tokens corresponding to each phoneme.

**Evaluation dataset.** We adopt the evaluation protocol proposed in VALL-E (Chen et al., 2025). Specifically, we filter the LibriSpeech test-clean subset to retain utterances ranging from 4 to 10 seconds in duration, resulting in a total of 2.2 hours of audio. For each utterance, a prompt is randomly sampled from another utterance spoken by the same speaker. A segment of 1, 3, or 5 seconds is extracted to serve as the prompt in our experiments.

### C.2 METRICS DETAILS

We assess each system utilizing the following objective evaluation metrics:

- **RTF** (Real-Time Factor) serves as a critical indicator of system efficiency, especially in applications that require real-time processing. It quantifies the duration needed to generate

---

**Algorithm 1** DiFlow-TTS Training

---

**Input**: Model $\mathcal{M}$, Dataset $\mathcal{D} = \{X^1, ..., X^M\}$, where each $X^i$ consists of phonemes $\mathbf{P}$, durations $\mathbf{d}$, ground-truth speech $\mathbf{y}$, and reference speech prompt $\mathbf{r}$.
**Output**: Trained Model $\mathcal{M}$

1: **while** $\mathcal{M}$ not converged **do**
2:     Sample $X \sim \mathcal{D}$
3:     Extract prosody tokens $\mathbf{y}^p$, content tokens $\mathbf{y}^c$, and acoustic tokens $\mathbf{y}^a$ from $\mathbf{y}$, as defined in Equation (2)
4:     Extract only prosody tokens $\mathbf{r}^p$, acoustic tokens $\mathbf{r}^a$, and speaker embedding $\mathbf{s}$ from $\mathbf{r}$, as defined Equation (2)
5:     $\mathbf{x}_1 \leftarrow \mathbf{y}^p \oplus \mathbf{y}^a$                                    ▷ Prosody + acoustic tokens as target
6:     $\mathbf{p} \leftarrow \text{PhonemeEncoder}(\mathbf{P})$
7:     $\hat{\mathbf{d}} \leftarrow \text{DurationPredictor}(\mathbf{p})$
8:     $\mathbf{p}_{\text{up}} \leftarrow \text{LengthRegulator}(\mathbf{p}, \mathbf{d})$
9:     $\mathbf{h} \leftarrow \text{ContentPredictor}(\mathbf{p}_{\text{up}})$
10:     Obtain $\mathbf{h}_c$ and $p(\cdot|\mathbf{h}; \varphi)$ as defined in Equation (3)
11:     $\mathcal{L}_{\text{dur}} \leftarrow \text{MSE}(\mathbf{d}, \hat{\mathbf{d}})$
12:     $\mathcal{L}_c \leftarrow \text{CE}(\mathbf{y}^c, p(\cdot|\mathbf{h}; \varphi))$
13:     Sample $t \sim \mathcal{U}(0, 1)$
14:     Sample $\mathbf{x}_t \sim p_{t|1}(\mathbf{x}_t \mid \mathbf{x}_1)$                                    ▷ Noising
15:     Obtain $\mathbf{h}_{p,a}$ using $\mathbf{x}_t, \mathbf{h}_c, \mathbf{r}^p, \mathbf{r}^a, \mathbf{s}, t$ as defined in Section 3 of the paper
16:     $\mathbf{h}_p, \mathbf{h}_a = \text{Slice}(\mathbf{h}_{p,a})$
17:     $p_{1|t}(\cdot|\mathbf{x}_t, \mathbf{c}; \theta) \leftarrow f_\phi(\mathbf{h}_p) \oplus f_\omega(\mathbf{h}_a)$                                    ▷ Denoising prediction
18:     $\mathcal{L}_{\text{FDFD}} \leftarrow \text{CE}(\mathbf{x}_1, p_{1|t}(\cdot|\mathbf{x}_t, \mathbf{c}; \theta))$
19:     $\mathcal{L} \leftarrow \lambda_{\text{dur}}\mathcal{L}_{\text{dur}} + \lambda_c\mathcal{L}_c + \lambda_{\text{FDFD}}\mathcal{L}_{\text{FDFD}}$
20:     Optimizer.step($\mathcal{L}$)
21: **end while**

---

one second of speech. RTF evaluations for all models are conducted in a complete end-to-end configuration on a single NVIDIA 80GB A100 GPU.

- **UTMOS** (Saeki et al., 2022) is a deep learning framework designed to gauge the naturalness and general quality of speech by estimating mean opinion scores (MOS). This approach mitigates the resource-intensive nature of traditional subjective assessments, using sophisticated neural networks to produce predictions that strongly correlate with human perceptual evaluations.

- **SIM-O** is a metric used to quantify the similarity of the speakers. It evaluates the resemblance between the synthesized speech and the original prompt. This metric is derived from the cosine similarity of the speaker embeddings obtained through WavLM-TDCNN[2] applied to the audio waveforms. SIM-O spans a range of -1 to 1, where higher values indicate stronger speaker similarity.

- **WER** (Word Error Rate) is utilized to appraise the robustness of speech synthesis systems, focusing on the precision of word pronunciation. An automatic speech recognition (ASR) model[3] transcribes the generated speech, which is then compared to the textual prompt. The employed ASR model is a connectionist temporal classification (CTC)-based HuBERT, pre-trained on LibriLight, and fine-tuned on the 960-hour LibriSpeech training dataset.

- **Prosody Accuracy & Error** metrics evaluate the congruence between the synthesized speech and the audio prompt, focusing on pitch (F0) and energy contours. Accuracy is determined following the framework outlined in PromptTTS (Guo et al., 2022) and TextrolSpeech (Ji et al., 2024b), by classifying F0 and energy into three tiers such as high, normal and low, which are relative to their mean values [4]. Furthermore, the Root Mean

---

[2] https://github.com/microsoft/UniSpeech/tree/main/downstreams/speaker_verification
[3] https://huggingface.co/facebook/hubert-large-ls960-ft
[4] https://github.com/jishengpeng/TextrolSpeech

---

**Algorithm 2** DiFlow-TTS Inference

---

**Input**: The phonemes $\mathbf{P}$, and reference speech prompt $\mathbf{r}$, the number of sampling step $N$, and step size $\Delta_t = \frac{1}{N}$.

**Output**: Synthesized speech $\hat{a}$.

1: Extract prosody tokens $\mathbf{r}^p$, acoustic tokens $\mathbf{r}^a$, and speaker embedding $\mathbf{s}$ from $\mathbf{r}$, as defined in Equation (2)
2: $\mathbf{p} \leftarrow \text{PhonemeEncoder}(\mathbf{P})$
3: $\hat{\mathbf{d}} \leftarrow \text{DurationPredictor}(\mathbf{p})$
4: $\mathbf{p}_{\text{up}} \leftarrow \text{LengthRegulator}(\mathbf{p}, \hat{\mathbf{d}})$
5: $\mathbf{h} \leftarrow \text{ContentPredictor}(\mathbf{p}_{\text{up}})$
6: Obtain $\mathbf{h}_c$ and $p(\cdot|\mathbf{h}; \varphi)$ using $\mathbf{h}$ as defined in Equation (3)
7: Sample $\mathbf{x}_0 \sim p(\mathbf{x}_0)$
8: **for** $t = 0$ **to** $1 - \Delta_t$ **with step** $\Delta_t$ **do**
9:     Obtain $\mathbf{h}_{p,a}$ using $\mathbf{x}_t$, $\mathbf{h}_c$, $\mathbf{r}^p$, $\mathbf{r}^a$, $\mathbf{s}$, $t$ as defined in Section 3 of the paper
10:    $\mathbf{h}_p, \mathbf{h}_a = \text{Slice}(\mathbf{h}_{p,a})$
11:    $p_{1|t}(\cdot|\mathbf{x}_t, \mathbf{c}; \theta) \leftarrow f_\phi(\mathbf{h}_p) \oplus f_\omega(\mathbf{h}_a)$       ▷ Denoising prediction
12:    Sample $\mathbf{x}_1^i \sim p_{1|t}^i(\cdot|\mathbf{x}_t, \mathbf{c}; \theta)$
13:    $\mathbf{u}_t^i(x^i|\mathbf{x}_t^i, \mathbf{x}_1^i) \leftarrow \frac{\dot{\kappa}_t}{1-\kappa_t}\left[\delta_{\mathbf{x}_1^i}(x^i) - \delta_{\mathbf{x}_t^i}(x^i)\right]$   ▷ Probability velocity as defined in Equation (1)
14:    $\lambda^i \leftarrow \sum_{x^i \neq \mathbf{x}_t^i} \mathbf{u}_t^i(x^i|\mathbf{x}_t^i, \mathbf{x}_1^i)$
15:    Sample $z^i \sim \mathcal{U}(0, 1)$
16:    **if** $z^i \leq 1 - \exp(-\Delta_t\lambda^i)$ **then**
17:        Sample $\mathbf{x}_{t+\Delta_t}^i \sim \frac{1}{\lambda^i}\mathbf{u}_t(\cdot|\mathbf{x}_t^i, \mathbf{x}_1^i)(1 - \delta_{\mathbf{x}_t^i}(\cdot))$ ▷ Transition to a new token; self-transitions are disallowed
18:    **else**
19:        Sample $\mathbf{x}_{t+\Delta_t}^i \sim \delta_{\mathbf{x}_t}(\cdot)$       ▷ No transition; retain current token
20:    **end if**
21: **end for**
22: $\mathbf{x}^p, \mathbf{x}^a = \text{Split}(\mathbf{x}_t)$
23: $\mathbf{x}^c \leftarrow \arg\max_x \text{softmax}(p(x \mid \mathbf{h}; \varphi))$
24: $\mathbf{x} = \mathbf{x}^p \oplus \mathbf{x}^c \oplus \mathbf{x}^a$
25: $\hat{a} \leftarrow \text{CodecDecoder}(\mathbf{x}, \mathbf{s})$

---

Square Error (RMSE) is calculated to measure deviations in F0 and the energy between the synthesized output and the reference prompts.

### C.3 BASELINES DETAILS

We compare our model with previous zero-shot TTS baselines, including:

- **VoiceCraft** (Peng et al., 2024) is a token infilling neural codec language model built on a Transformer decoder architecture, incorporating a two-step token rearrangement procedure that applies causal masking for bidirectional-context autoregressive generation and delayed stacking for multi-codebook efficiency, trained autoregressively with a loss function that weights earlier codebooks more heavily. We use the official code and the pre-trained checkpoint[5], trained on 9K hours of the GigaSpeech dataset (Chen et al., 2021).

- **F5-TTS** (Chen et al., 2024b) is a fully non-autoregressive (NAR) TTS system based on flow matching with DiT architecture, where text inputs are padded with filler tokens to align with speech lengths, bypassing the need for duration models, text encoders, or phoneme alignment. It contributes refinements to text representation using ConvNeXt (Liu et al., 2022) for better speech alignment and an inference-time Sway Sampling strategy that improves generation efficiency. We reproduce F5-TTS using the official code[6] and train it on 500 hours of the LibriTTS dataset.

---

[5] https://huggingface.co/pyp1/VoiceCraft/blob/main/830M_TTSEnhanced.pth
[6] https://github.com/SWivid/F5-TTS

- **NaturalSpeech 2** (Shen et al., 2024) is a latent diffusion model designed for zero-shot TTS, capable of generating high-fidelity audio from diverse text inputs. It utilizes a neural audio codec and a latent diffusion framework to produce natural-sounding speech and singing without requiring speaker-specific training data. We use the Amphion toolkit (Zhang et al., 2024) and the pre-trained weight[7], trained on 585 hours of the LibriTTS dataset.

- **VALL-E** (Chen et al., 2025) is a neural codec language model that treats TTS synthesis as a conditional language modeling task using discrete codes derived from an off-the-shelf neural audio codec, pre-trained on 60,000 hours of English speech data to enable in-context learning. We reproduce VALL-E using the Amphion toolkit[8](Zhang et al., 2024) and train it on 500 hours of the LibriTTS dataset.

- **OZSpeech** (Hieu et al., 2025) is a zero-shot TTS system that employs optimal transport conditional flow matching with one-step sampling, conditioned on a learned prior derived from disentangled, factorized speech components represented in token format to model individual attributes. It contributes a novel framework that bypasses traditional multi-step sampling processes by leveraging the learned prior for direct generation from text prompts, thereby reducing computational demands and enhancing precise attribute disentanglement in speech synthesis. We use the official code and the pre-trained checkpoint[9], trained on 500 hours of the LibriTTS dataset.

- **MaskGCT** (Wang et al., 2025c) is a fully NAR zero-shot TTS model structured as a two-stage generative codec transformer, where the first stage predicts semantic tokens from input text using representations from a speech self-supervised learning model, and the second stage generates acoustic tokens conditioned on these semantic tokens via a mask-and-predict paradigm. It contributes an efficient training approach that learns to infill masked tokens based on prompts and conditions, enabling parallel inference for tokens of arbitrary length without explicit text-speech alignment or phone-level duration modeling, thus resolving key limitations in prior autoregressive and NAR TTS frameworks. We use the official code and the pretrained checkpoint [10], trained on English and Chinese data from Emilia (He et al., 2024), each with 50K hours of speech (totaling 100K hours). Since the baseline requires the total speech length, we use the ground-truth duration during inference.

## D ADDITIONAL ANALYSIS

**Prompt Duration.** To investigate in detail the influence of prompt duration on zero-shot speech synthesis, we conducted a comprehensive evaluation of DiFlow-TTS with 128 NFE across different prompt lengths: 1 second, 3 seconds (as reported in the paper), and 5 seconds. As shown in Table 6, increasing the prompt duration consistently improves all aspects of speech quality across models. Specifically, DiFlow-TTS, along with OZSpeech, achieves the lowest WER across all prompt lengths, demonstrating superior content preservation. In terms of naturalness and overall quality, our method attains SOTA performance, achieving the highest UTMOS score (4.00) with a 5-second prompt. Notably, this is achieved using only 470 hours of training data, whereas VoiceCraft (9K hours) and MaskGCT (100K hours) obtain lower UTMOS scores of 3.58 and 3.89, respectively. For speaker similarity, our method does not show a clear advantage over baseline models, though we note that these baselines also exhibit trade-offs under limited training data. Regarding pitch and energy accuracies and errors, which reflect prosody reconstruction ability, DiFlow-TTS consistently ranks as the best or second-best performer across prompt lengths. Overall, DiFlow-TTS strikes a favorable balance among naturalness, prosody, model size, and speaker similarity.

---

[7]https://huggingface.co/amphion/naturalspeech2_libritts/tree/main/checkpoint
[8]https://github.com/open-mmlab/Amphion
[9]https://github.com/ozspeech/OZSpeech
[10]https://huggingface.co/amphion/MaskGCT

| Type | Model | Data (hours) | UTMOS ↑ | WER ↓ | SIM-O ↑ | F0 Accuracy (↑) | F0 RMSE ↓ | Energy Accuracy ↑ | Energy RMSE ↓ |
|---|---|---|---|---|---|---|---|---|---|
| - | Ground Truth | - | 4.10 | 0.02 | - | - | - | - | - |
| *1s Prompt* | | | | | | | | | |
| (i) | VoiceCraft [†] | GS (9K) | 3.45 | 0.16 | 0.31 | 0.61 | 31.57 | 0.52 | **0.010** |
| | VALL-E [◇] | LT (500) | 3.61 | 0.21 | 0.24 | 0.55 | 37.87 | 0.40 | 0.020 |
| (ii) | NaturalSpeech 2 [‡] | LT (585) | 2.12 | 0.12 | 0.20 | 0.69 | 26.48 | 0.39 | 0.020 |
| | F5-TTS [◇] | LT (500) | 3.73 | 0.19 | 0.32 | 0.61 | 29.93 | 0.50 | 0.020 |
| | OZSpeech [†] | LT (500) | 3.17 | **0.05** | 0.30 | 0.62 | 27.70 | 0.49 | 0.020 |
| (iii) | MaskGCT [†] | E (100K) | 3.60 | 0.10 | **0.36** | 0.63 | 29.63 | **0.60** | 0.013 |
| (iv) | **DiFlow-TTS** | **LT (470)** | **3.74** | **0.05** | 0.34 | **0.82** | **13.00** | 0.55 | **0.010** |
| *3s Prompt* | | | | | | | | | |
| (i) | VoiceCraft [†] | GS (9K) | 3.55 | 0.18 | 0.51 | 0.78 | 17.22 | 0.44 | 0.010 |
| | VALL-E [◇] | LT (500) | 3.68 | 0.19 | 0.40 | 0.75 | 21.66 | 0.36 | 0.020 |
| (ii) | NaturalSpeech 2 [‡] | LT (585) | 2.38 | 0.09 | 0.31 | 0.80 | 15.62 | 0.25 | 0.020 |
| | F5-TTS [◇] | LT (500) | 3.76 | 0.24 | 0.52 | 0.80 | 13.78 | 0.67 | 0.010 |
| | OZSpeech [†] | LT (500) | 3.15 | **0.05** | 0.40 | 0.81 | 11.96 | 0.67 | 0.010 |
| (iii) | MaskGCT [†] | E (100K) | 3.83 | 0.09 | **0.67** | 0.77 | 14.33 | **0.75** | **0.007** |
| (iv) | **DiFlow-TTS** | **LT (470)** | **3.98** | **0.05** | 0.45 | **0.88** | **7.97** | 0.73 | **0.007** |
| *5s Prompt* | | | | | | | | | |
| (i) | VoiceCraft [†] | GS (9K) | 3.58 | 0.19 | 0.56 | 0.81 | 14.48 | 0.46 | 0.010 |
| | VALL-E [◇] | LT (500) | 3.72 | 0.19 | 0.46 | 0.79 | 18.20 | 0.41 | 0.010 |
| (ii) | NaturalSpeech 2 [‡] | LT (585) | 2.33 | 0.09 | 0.35 | 0.84 | 13.13 | 0.28 | 0.020 |
| | F5-TTS [◇] | LT (500) | 3.71 | 0.32 | 0.57 | 0.83 | 11.20 | 0.68 | 0.010 |
| | OZSpeech [†] | LT (500) | 3.15 | **0.05** | 0.39 | 0.83 | 12.05 | 0.67 | 0.010 |
| (iii) | MaskGCT [†] | E (100K) | 3.89 | 0.09 | **0.74** | 0.81 | 11.82 | **0.77** | 0.005 |
| (iv) | **DiFlow-TTS** | **LT (470)** | **4.00** | **0.05** | 0.48 | **0.89** | **8.04** | 0.73 | **0.007** |

Table 6: Performance on the *LibriSpeech test-clean* dataset across different audio prompt lengths. [◇] means reproduced results. [†] and [‡] mean results inferred from official and unofficial checkpoints, respectively. The best and second best are **bold** and underlined. Abbreviation: E (Emilia), GS (GigaSpeech), LT (LibriTTS).

