# OpenReview forum: "DiFlow-TTS: Compact and Low-Latency Zero-Shot Text-to-Speech with Factorized Discrete Flow Matching"
_ICLR.cc/2026/Conference — ICLR 2026 Conference Withdrawn Submission_

### Official Review · Reviewer_Hre9 · 2025-10-26

**Soundness:** 1
**Presentation:** 1
**Contribution:** 2
**Rating:** 2
**Confidence:** 5

**Summary:**

This paper introduces DiFlow-TTS, a zero-shot TTS system applying discrete flow matching on FACodec tokens. In specific, the paper leverages a Phoneme-Content Mapper for accurate semantic alignment and a Factorized Discrete Flow Denoiser that uses separate prediction heads to explicitly model distinct speech aspects. Experimental results demonstrate that DiFlow-TTS achieves state-of-the-art performance in naturalness, content accuracy, and prosody preservation while maintaining a significantly smaller model size (up to 11.7x smaller) and achieving substantially lower inference latency (up to 34x faster) than competing baselines.

**Strengths:**

1.The paper is the first to use discrete flow matching for zero-shot TTS

2.The paper achieves enhanced quality with smaller model size and faster inference speed.

3.The code is released.

**Weaknesses:**

1.The paper's central novelty lies in applying a discrete generative model to factorized tokens. However, this general approach has been explored using discrete diffusion models, most notably in NaturalSpeech 3 which introduced the FACodec. To better isolate the contribution of using discrete flow matching(DFM), the paper would be significantly strengthened by: (1) explicitly discussing the theoretical and practical differences between DFM and discrete diffusion in the methodology section, and (2) including an experimental baseline using a discrete diffusion model trained on the same FACodec tokens to provide a direct comparison of the two generative frameworks.

2.A notable weakness is the model's performance on speaker similarity, as measured by SIM-O, which lags considerably behind state-of-the-art models like MaskGCT. Given that high-fidelity voice cloning is a central goal of zero-shot TTS, this performance gap is significant.

3.There is an inconsistency in the paper's framing. The authors criticize prior works for the "redundant" step of embedding discrete inputs to a continuous space (L55-57), claiming their "direct discrete formulation removes this conversion." However, their own method employs "prosody and acoustic embedders" (L236), which performs an analogous function. The manuscript's claims should be revised to reflect this more precise and accurate distinction.

4.The assertion that continuous representations are inherently more prone to attribute entanglement than discrete ones (Abstract and L51) is a strong claim that is not empirically substantiated within the paper. To validate this claim, a comparative experiment modeling the continuous embeddings from the VQ-VAE codebooks in FACodec would be necessary. Otherwise, the authors should moderate this claim.

**Questions:**

See weaknesses.

---

> ### Author Response · Authors · 2025-11-27
> **Response to Reviewer Hre9 [1/2]**
>
> **[W2]**: Performance on speaker similarity
>
> Since our study aims to investigate the viability of DFM for speech synthesis, we adopt FaCodec because it factorizes speech into prosody, content, and acoustic attributes, making these components easier to model and control. Although our proposed method significantly improves the quality of generated speech, it still faces certain limitations. Particularly, our framework relies on FaCodec, which disentangles complex speech attributes and speaker identity, integrating it into any framework requires additional mechanisms to effectively inject speaker information. We acknowledge that our current strategy for this injection is not yet optimal. In contrast, codecs such as EnCodec inherently contain speaker information within their vector quantizers, allowing the model to implicitly learn speaker characteristics. Furthermore, SIM-O is an embedding-based objective metric, which may overlook perceptual cues that humans rely on to judge speaker similarity. This is consistent with our results: although our SIM-O score is lower than some baselines, our model achieves a top score in the subjective MOS evaluation of speaker similarity. While the objective SIM-O metric remains lower than several state-of-the-art baselines, our key contribution lies in achieving consistent improvements across multiple metrics, along with strong human-perceived similarity.
>
> ---
> **[W3]**: Inconsistency in the paper's framing
>
> We thank the reviewer for this insightful comment. It is indeed true that our manuscript (Lines 55–57) mentions that prior works often perform a "redundant" conversion by embedding discrete inputs into a continuous space before applying continuous flow matching. Our intention was to emphasize that our method adopts a discrete generative framework, specifically discrete flow matching, which directly models transitions between discrete tokens rather than relying on a continuous latent denoising process.
>
> We acknowledge that the "prosody and acoustic embedders" (Line 236) introduce an embedding step; however, these modules serve a different role. They act as feature encoders for conditioning rather than as part of a continuous noising and denoising latent process typical in continuous flow pipelines. In our framework, the core generative mechanism operates directly over discrete codec tokens, while the embedding simply maps input features.
>
> To clarify this point, we will revise the manuscript to explicitly state that "Our approach removes the need to represent discrete tokens as continuous latent variables within the generative flow matching process".

---

> ### Author Response · Authors · 2025-11-28
> **Response to Reviewer Hre9 [2/2]**
>
> **[W1]** Clarification of novelty
>
> We appreciate the constructive feedback. Due to time and computational constraints during the rebuttal phase, we were unable to allocate sufficient resources to complete these additional experiments. Instead, we address the reviewer's concerns by (1) explicitly clarifying the theoretical and practical distinctions between DFM and discrete diffusion, and (2) highlighting the novelty and contributions of DiFlow-TTS.
>
> 1. **Differences between DFM and Discrete Diffusion**
>
> We acknowledge that discrete diffusion frameworks have recently shown strong performance in speech synthesis. However, diffusion-based methods inherently couple the training and sampling processes, restricting sampling configurations to those established during training [1]. Consequently, modifying components such as noise schedules or rate matrices requires retraining for each new configuration, resulting in considerable computational overhead. In contrast, DFM performs the noising process through linear interpolation in probability space between clean and noisy data, allowing sampling with arbitrary step sizes and any number of steps while remaining fully decoupled from the training procedure. This decoupling enables flexible and efficient sampling that can be optimized independently of training, avoiding costly retraining. While these properties make DFM a promising generative paradigm, its application to speech synthesis remains underexplored, and how to best leverage this flexibility in practice is still an open question.
>
> 2. **Novelty and justification of DiFlow-TTS**
>
> DFM has well-established advantages over discrete diffusion frameworks, as demonstrated in prior studies comparing the two generative paradigms, particularly in terms of training efficiency, sampling flexibility, and overall performance [1]. While factorized discrete codes have previously been combined with diffusion models, integrating DFM into the zero-shot TTS domain introduces a fundamentally different formulation. This is a key novelty of DiFlow-TTS. Bridging DFM with a factorized codec is not a simple substitution; it required designing a deterministic content modeling module (e.g., PCM) and a tailored factorized flow architecture capable of jointly handling multiple discrete code streams (e.g., prosody and acoustic) within a single model, rather than training separate diffusion models for each attribute. This design not only simplifies the model structure, reducing the number of submodules, but also lowers the overall parameter count. The success of DiFlow-TTS demonstrates that DFM is a viable and advantageous alternative to diffusion for this task, a direction that has not been explored previously. In doing so, our work highlights that DFM can achieve comparable or even superior results with substantially greater efficiency. This contribution remains distinct even if the general concept of discrete generative modeling on factorized tokens exists, since the underlying generative framework, flow versus diffusion, fundamentally impacts both system complexity and performance.
>
> ---
> **[W4]** Entanglement on continuous representations
>
> We thank the reviewer for raising this important point. We agree that the statement regarding continuous representations being more susceptible to attribute entanglement is strong and should be moderated. We acknowledge that the current manuscript does not include a direct empirical comparison using the continuous embeddings from the VQ-VAE codebooks in FaCodec, and due to time and computational constraints during the rebuttal phase, we are unable to include such an experiment. However, we have observed similar behavior in our experiments. While this evidence is not definitive, it provides useful intuition supporting our claim. Specifically, we offer indirect evidence from a baseline that relies on continuous latent modeling with FaCodec, namely OZSpeech. OZSpeech uses an intermediate encoder to map FaCodec attributes into a continuous latent space before applying continuous flow matching. As shown in Table 1, OZSpeech lags considerably behind our model across all key metrics. This performance gap suggests that continuous latent modeling may introduce attribute blending effects that impact downstream synthesis quality.
>
> To ensure accuracy, we will revise the manuscript to moderate the claim and clarify that our observations are informed by trends reported in prior work and indirect evidence from continuous-latent baselines such as OZSpeech.
>
> [1] Qin, Yiming, et al. "DeFoG: Discrete Flow Matching for Graph Generation", ICML (2025)

---

> > ### Author Response · Authors · 2025-11-28
> > **Official Comment by Authors**
> >
> > Dear Reviewer,
> >
> > We hope our responses have addressed your concerns. Please let us know if any additional clarification is needed. We truly value your feedback.
> >
> > Best regards,
> >
> > The Authors

---

### Official Review · Reviewer_bruE · 2025-10-29

**Soundness:** 3
**Presentation:** 4
**Contribution:** 2
**Rating:** 2
**Confidence:** 3

**Summary:**

This paper presents DiFlow-TTS, a non-autoregressive, zero-shot text-to-speech system that introduces a novel approach by leveraging purely Discrete Flow Matching (DFM). The core idea is to model the generation process directly in the discrete space of factorized speech tokens, avoiding the common practice of mapping them to a continuous space. The architecture is composed of a speech tokenizer that disentangles audio into prosody, content, and acoustic attributes; a Phoneme-Content Mapper for robust text alignment; and the central Factorized Discrete Flow Denoiser (FDFD). The FDFD uses a Diffusion Transformer (DiT) architecture with dedicated prediction heads to explicitly model the distributions of prosody and acoustic details. The authors provide a thorough empirical evaluation demonstrating that DiFlow-TTS achieves state-of-the-art performance while being substantially smaller and faster than a range of baseline models.

**Strengths:**

1. Novel and Well-Motivated Application of DFM: The primary strength of this work lies in its innovative application of a purely discrete generative framework to speech synthesis. By demonstrating the viability of DFM directly on factorized speech tokens, the paper presents a principled and elegant alternative to continuous diffusion models, simplifying the generation pipeline and reducing computational overhead. This is a timely and significant contribution to the field.

2. Excellent Performance and Efficiency Balance: The empirical results are very strong. The paper shows that DiFlow-TTS achieves top-tier performance when compared against the chosen baselines. This high quality is achieved with remarkable efficiency. The model offers much lower latency than competitors, establishing a very favorable quality-efficiency trade-off.

3. Rigorous Experimental Validation: The paper is supported by a comprehensive and well-designed set of experiments. The authors benchmark against a diverse suite of models and employ a wide range of metrics, including objective, subjective, and prosody-specific evaluations. The ablation studies are particularly effective, systematically confirming the importance of the key design choices, such as the factorized prediction heads and the inclusion of various conditioning signals.

**Weaknesses:**

1. Limited Competitiveness of Baselines: The primary weakness of this paper is that the selected set of baselines may not represent the current state-of-the-art in text-to-speech synthesis. While models like VALL-E and NaturalSpeech 2 were foundational, the field has seen rapid progress with the advent of powerful, large-language-model-based systems. The omission of more recent and stronger competitors makes it difficult to accurately situate DiFlow-TTS's performance. This weakens the central claim of superiority, as the reported performance gap might be significantly smaller when compared against true state-of-the-art systems.

2. Discrepancy in Speaker Similarity Metrics: While the model achieves a top score in the subjective evaluation of speaker similarity (MOS), the objective SIM-O metric remains noticeably lower than that of several key baselines (e.g., F5-TTS, MaskGCT).

3. Reliance on Pre-Existing Representations Constrains Novelty: The paper's core innovation lies in applying DFM to a pre-existing, factorized token representation from FACodec. While this is a valid and effective approach, it frames the work's novelty as being primarily in the generative methodology rather than in the end-to-end synthesis pipeline or the speech representation itself. The impressive performance is therefore a joint product of the proposed DFM model and the powerful capabilities of the external codec, which can make it difficult to isolate the precise contribution of the flow matching component to the final quality.

**Questions:**

1. The comparison with OZSpeech is crucial for evaluating the paper's claims on efficiency. Table 4 reveals that DiFlow-TTS achieves a UTMOS of 2.904 with a single function evaluation (NFE=1). In contrast, Table 3 shows that OZSpeech, a continuous flow-matching model, reaches a higher UTMOS of 3.15 in the same single-step setting. This suggests that at the lowest possible latency (NFE=1), the continuous flow matching approach currently outperforms the proposed discrete method in terms of speech naturalness.

To make a complete assessment, could the authors provide a full evaluation of DiFlow-TTS at NFE=1, including all key metrics from Table 1 (WER, SIM-O, F0/Energy RMSE)? Such a comparison is critical for justifying the central motivation of the paper. If the discrete flow matching approach does not demonstrate a clear advantage at its most efficient, single-step inference point, what is the compelling reason to prefer it over established continuous flow matching methods for low-latency TTS?

2. The field of generative speech is advancing very rapidly. The current baseline set is strong, but have you considered benchmarking against more recent and powerful LLM-based systems like CosyVoice 2 [1] or Spark-TTS [2]? Including them, even if trained on larger datasets, would provide a more complete picture of the current state of the art.

3. The ablation study effectively demonstrates the importance of attribute-type embeddings. Would it be possible to conduct a finer-grained study by removing only the prosody attribute embedding or only the acoustic attribute embedding? This could provide deeper insights into how each attribute-specific signal contributes to the final synthesis quality.

[1] Du, Zhihao, et al. "Cosyvoice 2: Scalable streaming speech synthesis with large language models." arXiv preprint arXiv:2412.10117 (2024).
[2] Wang, Xinsheng, et al. "Spark-tts: An efficient llm-based text-to-speech model with single-stream decoupled speech tokens." arXiv preprint arXiv:2503.01710 (2025).

---

> ### Author Response · Authors · 2025-11-27
> **Response to Reviewer bruE [1/2]**
>
> **[W1, Q2]**: Limited competitiveness of baselines
>
> We appreciate your concern and agree that the field of generative speech synthesis is advancing rapidly, with several recent TTS systems achieving impressive performance. The baselines used in our study are drawn from works published between 2024 and 2025, and therefore remain relevant and representative of recent progress in the field. While we have considered including stronger baselines, many of these models are trained on **extremely large-scale datasets** containing tens to hundreds of thousands of hours of speech. Although such systems naturally achieve higher performance, the substantial difference in data scale makes direct comparison less fair within the limited-data setting of our work. Our goal is to evaluate DiFlow-TTS under realistic low-resource conditions and compare it against baselines trained on a comparable scale, while also highlighting cases where large-scale models still underperform ours. For example, DiFlow-TTS outperforms the recent MaskGCT baseline across several key metrics, even though MaskGCT is trained on 100K hours of speech, demonstrating the strong data efficiency of our approach. Due to time and computational constraints during the rebuttal phase, retraining on massive datasets is not feasible; however, a scaled-up version with larger model capacity and broader data coverage will be released in future work.
>
> Following the your suggestion, we additionally report results against CosyVoice 2 and Spark-TTS. As shown in the table, DiFlow-TTS achieves lower WER and stronger prosody-related metrics than both systems, further validating the effectiveness of our approach despite the substantial differences in training data scale.
>
> | Model       | Data (hours) | WER (%) | UTMOS | SIM-O | F0 ACC | F0 RMSE | Energy ACC | Energy RMSE |
> |-------------|--------------|--------|----------|----------|------------|--------------|----------------|------------------|
> | CosyVoice2  | 30K          | 9.32   | 2.75     | 0.12     | 0.34       | 56.16        | 0.67           | 0.008            |
> | Spark-TTS   | 100K         | 9.79   | 4.31     | 0.57     | 0.87       | 10.02        | 0.52           | 0.014            |
> | DiFlow-TTS  | 470          | 4.83   | 3.98     | 0.45     | 0.88       | 7.97         | 0.73           | 0.007            |
>
> ---
> **[W3]** Reliance on pre-existing representations constrains novelty
>
> Our work, as the first discrete flow matching–based speech synthesis model, introduces advances that extend beyond existing generative frameworks and offer practical innovations for speech synthesis. DiFlow-TTS leverages the principles of DFM to address a clear gap in the speech generation landscape, where discrete diffusion has been explored but DFM has not. We appreciate your point and agree that FACodec provides a strong factorized representation. However, our contribution is orthogonal to the codec design. Our aim is not to propose a new speech representation, but to develop a new generative mechanism for modeling discrete speech tokens. If the codec were the dominant factor in performance, then any model operating on FACodec tokens should achieve comparable results. This is contradicted by empirical evidence (Table 1): OZSpeech, which also operates on FACodec tokens, exhibits significantly weaker performance. This discrepancy highlights that the generative framework, rather than the codec alone, plays a central role in determining speech quality.

---

> ### Author Response · Authors · 2025-11-27
> **Response to Reviewer bruE [2/2]**
>
> **[W2]**: Discrepancy in speaker similarity metrics
>
> Our study aims to investigate the viability of DFM for speech synthesis. We adopt FaCodec because it factorizes speech into prosody, content, and acoustic attributes, making these components easier to model and control. Although our proposed method significantly improves the quality of generated speech, it still faces certain limitations. Since our framework relies on the third-party FaCodec, which disentangles complex speech attributes and speaker identity, integrating it into any framework requires additional mechanisms to effectively inject speaker information. We acknowledge that our current strategy for this injection is not yet optimal. In contrast, codecs such as EnCodec inherently contains speaker information within their vector quantizers, allowing the model to implicitly learn speaker characteristics. Furthermore, the SIM-O metric, being embedding-based, may overlook perceptual attributes that are important to human listeners. While our model may not show a clear advantage over baselines in terms of SIM-O, our key contribution lies in achieving consistent improvements across multiple metrics as well as strong performance in human evaluations, rather than focusing solely on SIM-O.
>
> ---
> **[Q1]**: The comparison with OZSpeech
>
> Our work, as the first discrete flow matching–based speech synthesis model, introduces advances that extend beyond existing generative frameworks and offer practical innovations for speech synthesis. DiFlow-TTS leverages the principles of DFM to address a clear gap in the speech generation landscape, where discrete diffusion has been explored but DFM has not. We appreciate the reviewer’s point and agree that FACodec provides a strong factorized representation. However, our contribution is orthogonal to the codec design. Our aim is not to propose a new speech representation, but to develop a new generative mechanism for modeling discrete speech tokens. If the codec were the dominant factor in performance, then any model operating on FACodec tokens should achieve comparable results. This is contradicted by empirical evidence (Table 1): OZSpeech, which also operates on FACodec tokens, exhibits significantly weaker performance. This discrepancy highlights that the generative framework, rather than the codec alone, plays a central role in determining speech quality.
>
> ---
> **[Q3]**: Finer-grained ablation study
>
> We have followed your suggestion and conducted a finer-grained ablation to isolate the contribution of each attribute-specific embedding. The additional results are reported in the table below. We observe that removing either the prosody attribute embedding or the acoustic attribute embedding leads to noticeable performance degradation, particularly in naturalness and metrics closely tied to these attributes. Attribute-type embeddings guide the model in distinguishing which attribute each token corresponds to when feeding them into the DiT. When an attribute embedding is removed, the model is forced to treat all attributes uniformly, which results in entanglement and reduced synthesis quality.
> | Model                             | WER (%)  | UTMOS | SIM-O | F0 ACC | F0 RMSE | Energy ACC | Energy RMSE |
> |----------------------------------|------|-------|--------|--------|---------|-------------|--------------|
> | w/o Prosody Attribute Embedding  | 6.17 | 3.918 | 0.445  | 0.846  | 9.547   | 0.734       | 0.007        |
> | w/o Acoustic Attribute Embedding | 5.96 | 3.929 | 0.447  | 0.868  | 9.216   | 0.741       | 0.007        |
> | DiFlow-TTS                       | 4.83 | 3.978 | 0.454  | 0.884  | 7.972   | 0.735       | 0.007        |

---

> > ### Author Response · Authors · 2025-11-28
> > **Official Comment by Authors**
> >
> > Dear Reviewer,
> >
> > We hope our responses have addressed your concerns. Please let us know if any additional clarification is needed. We truly value your feedback.
> >
> > Best regards,
> >
> > The Authors

---

### Official Review · Reviewer_4aXh · 2025-10-31

**Soundness:** 3
**Presentation:** 3
**Contribution:** 2
**Rating:** 4
**Confidence:** 4

**Summary:**

This paper introduces DiFlow-TTS, a zero-shot text-to-speech framework that performs flow matching directly in the discrete space of factorized codec tokens, avoiding the continuous embeddings used in prior “discrete” flow-based methods. The model leverages a Phoneme-Content Mapper (PCM) to align phoneme sequences with discrete speech tokens for semantic grounding, and a Factorized Discrete Flow Denoiser (FDFD) with separate prediction heads for prosody and acoustic details, enabling explicit modeling of aspect-specific distributions. Experimental results show that DiFlow-TTS achieves strong speech quality, naturalness, and prosody preservation, while maintaining a compact model (up to 11.7× smaller) and low-latency inference (up to 34× faster) than state-of-the-art baselines. The approach provides an efficient and controllable discrete generative framework for zero-shot speech synthesis.

**Strengths:**

S1:The paper explores discrete flow matching (DFM) for zero-shot TTS — an area still dominated by diffusion- or AR-based models. Applying DFM directly on factorized codec tokens (prosody, content, acoustic) without continuous embeddings is novel in this context. The proposed Factorized Discrete Flow Denoiser (FDFD) and Phoneme-Content Mapper (PCM) together form a coherent pipeline that cleanly disentangles speech attributes.

S2:The method is generally well-formulated, and the paper carefully adapts the discrete flow formulation (Gat et al., 2024) to token-based speech synthesis. The derivations are mathematically consistent, and most architectural details are explicit. Ablation studies are informative and show that each component contributes meaningfully (Table 5). DiFlow-TTS shows competitive or superior naturalness (UTMOS/MOS) compared to strong baselines like F5-TTS and MaskGCT, with a compact 164M parameter size.

**Weaknesses:**

W1: The definition of the probability velocity in Eq. (1) appears to directly subtract a Kronecker delta term, rather than the marginal path distribution $p_t(x^i \mid x_t)$. While this form is mathematically consistent with Gat et al. (2024, Eq. 24) under the deterministic convex-interpolant path assumption (i.e., $p_t(x^i \mid x_t) = \delta_{x_t}(x^i)$), it would improve clarity if the authors explicitly stated this assumption. Otherwise, readers might interpret the formula as omitting $p_t$ from the more general flow-matching definition.

W2: The proposed Phoneme-Content Mapper (PCM) appears to follow the same structure as conventional duration-based alignment modules used in models such as FastSpeech2, consisting of a phoneme encoder, duration predictor, length regulator, and feed-forward Transformer layers. While it serves an important role in aligning textual phonemes with discrete codec tokens (which differ from mel frames), the architecture itself is not novel. I suggest clarifying that PCM is an adaptation of existing duration-alignment mechanisms to discrete token modeling, rather than a core methodological contribution.

W3:The formulation in Eq. (3) is consistent in terms of dimensionality and model structure.
The projection branch $\mathcal{H}\rho(\cdot)$ outputs content embeddings
$\mathbf{h}c \in \mathbb{R}^{n\times L\times D}$,
and the content head $\mathcal{G}\varphi(\cdot)$ produces logits over a vocabulary of size $v$,
$\mathcal{G}\varphi(\mathbf{h}) \in \mathbb{R}^{n\times L\times v}$.
However, the notation
$p(x^c \mid \mathbf{h}; \varphi) = \mathcal{G}\varphi(\mathbf{h})$
is slightly imprecise: since $\mathcal{G}\varphi(\cdot)$ outputs unnormalized logits rather than a normalized probability distribution,
a more accurate form would be
$p(x^c \mid \mathbf{h}; \varphi) = \text{softmax}(\mathcal{G}_\varphi(\mathbf{h}))$.

W4:The equation in section 3.4 is mathematically consistent, but the accompanying text is misleading: the embeddings are concatenated along the quantizer (attribute) dimension, not the temporal one. The shape $(m+n+k)\times(L_p+L)\times D$ confirms this.

W5:WER is often reported as a fraction (e.g., 0.05 = 5%), but this is not stated in the table. Please clarify the unit, as readers may misinterpret the numbers otherwise.

W6: The comparison with OZSpeech (145 M params, NFE = 1, RTF = 0.03, WER = 0.05) suggests that DiFlow-TTS does not exhibit a clear advantage in speed or word-error performance at a similar model scale. While the UTMOS score is slightly higher, the difference appears modest given the comparable inference cost. I encourage the authors to clarify what specific architectural or training differences contribute to the claimed efficiency gains.

**Questions:**

Q1:Could the authors include comparisons against other recent zero-shot TTS models (e.g., NaturalSpeech 3 (Ju et al., 2024), E2 TTS (Eskimez et al., 2024), ZipVoice (Zhu et al., 2025), VALL-E 2 (Chen et al., 2024)) in terms of model size, inference steps, RTF and WER? This would strengthen the claim that the proposed method offers a distinct speed-quality trade-off.
Q2: The reported WER values for several baselines (e.g., VALL-E and NaturalSpeech) differ by nearly an order of magnitude from those reported in other papers such as NaturalSpeech 3 (e.g., VALL-E ≈ 6 % vs. 0.19 here). Even if we ignore unit scaling, the absolute values seem substantially lower. Could the authors clarify which ASR system was used to compute WER and whether their evaluation follows the same protocol as previous works. Using a consistent WER computation method would make cross-paper comparisons more meaningful.
Q3: The paper’s core motivation is that discrete, factorized representations enable better attribute disentanglement (prosody, content, and acoustic detail) compared to continuous embeddings. While Table 5 and the Effect of Each Component analysis show that removing individual heads degrades perceptual metrics, these results only confirm that each branch contributes to quality rather than directly validating disentanglement. Additional evidence or analyses would be helpful to substantiate this key motivation.

---

> ### Author Response · Authors · 2025-11-26
> **Response to Reviewer 4aXh [1/2]**
>
> **[W1 - W5]** Thank you for your valuable feedback, which significantly helps us improve the clarity and quality of the manuscript. We agree with all of your comments and acknowledge that these issues stemmed from oversights in our writing. We will revise the manuscript accordingly, addressing the following points:
> - Explicitly stating the deterministic convex-interpolant assumption in Eq. (1).
> - Clarifying that PCM is an adaptation of existing duration-alignment mechanisms for discrete tokens.
> - Correcting the notation for unnormalized logits in Eq. (3).
> - Fixing the description of the embedding concatenation dimension.
> - Clearly indicating that WER is reported as a fraction.
> ---
> **[W6]**: The comparison with OZSpeech
>
> We agree that OZSpeech achieves strong one-step performance because it is **specifically trained as a single-step model** and only supports one-step inference. During training, it introduces **additional prior and anchor losses** that explicitly encourage reducing the number of sampling steps. Therefore, it is expected to outperform our model in the strictly one-step setting. However, our goal is different: DiFlow-TTS is designed to support flexible multi-step sampling. In the manuscript, we emphasize that DiFlow-TTS achieves competitive one-step performance **without requiring any extra losses**, while also enabling higher-quality generation with only a small number of sampling steps.

---

> ### Author Response · Authors · 2025-11-26
> **Response to Reviewer 4aXh [2/2]**
>
> **[Q1]**: Comparisons against other recent zero-shot TTS models
>
> We have followed your request and report the additional results in the table below. For NaturalSpeech 3, only the FACodec component is publicly available, and the full TTS model is not released for benchmarking, so we are unable to evaluate it.
>
> | Model              | Data (hours) | NFE | Model Size | RTF  | WER (%)  | UTMOS | F0 ACC | F0 RMSE | Energy ACC | Energy RMSE |
> |--------------------|--------------|-----|------------|------|------|--------|--------|----------|-------------|--------------|
> | CosyVoice2         | 30K          | 10  | 618M       | 0.40 | 9.32 | 2.75   | 0.34   | 56.16    | 0.67        | 0.008        |
> | E2 TTS             | 50K          | 32  | 333M       | 0.18 | 4.24 | 3.54   | 0.82   | 12.54    | 0.66        | 0.010        |
> | ZipVoice           | 100K         | 16  | 123M       | 0.11 | 3.93 | 4.05   | 0.81   | 14.13    | 0.64        | 0.014        |
> | ZipVoice-Distill   | 100K         | 8   | 123M       | 0.06 | 3.52 | 4.16   | 0.79   | 13.74    | 0.63        | 0.014        |
> | DiFlow-TTS    | 470      | 16  | 122M       | 0.05 | 5.46 | 3.89   | 0.87   | 8.58     | 0.73        | 0.008        |
>
> As shown in the table, the comparison demonstrates that DiFlow-TTS delivers competitive performance on several key metrics, even though it is trained on **only 470 hours of speech**, which is **roughly 200 times smaller** than the datasets used by recent large-scale models such as CosyVoice2, E2 TTS, and ZipVoice. Notably, DiFlow-TTS achieves an RTF of 0.05, which is competitive with the reported RTFs of these models, while maintaining reasonable WER, naturalness, and prosody-related metrics. This highlights that DFM is an effective choice for designing low-latency TTS models.
>
> ---
> **[Q2]**: Clarification on WER evaluation protocol
>
> We acknowledge that the WER values for several baselines differ from those reported in NaturalSpeech 3. This discrepancy arises because NaturalSpeech 3 re-trained these baselines on the **LibriLight 60k-hour dataset**, which is more than **100 times larger** than the dataset used in our experiments. In contrast, to maintain a fair comparison within our experimental constraints, we reproduced all baselines using **only the 500-hour LibriTTS subset** employed in our study (see the “Data (hours)” column in Table 1). As a result, the absolute WER values are not expected to match those reported in large-scale reproductions.
>
> For the ASR system, we adhered to the same evaluation protocol used in NaturalSpeech 3. We employed a CTC-based HuBERT ASR model pre-trained on LibriLight and fine-tuned on the 960-hour LibriSpeech training set, as detailed in Appendix C.2.
>
> ---
> **[Q3]**: Clarification of disentanglement motivation
>
> Our study aims to investigate the viability of DFM for speech synthesis. As the first discrete flow-matching–based speech synthesis model, we adopt FACodec because it factorizes speech into prosody, content, and acoustic attributes, making these components easier to model and control. We clarify that our goal is to preserve this factorized structure when integrating it into a generative framework. As shown in the last row of Table 5, replacing the multi-head prediction with a single head causes the model to implicitly learn the joint distribution of prosody and acoustic attributes, which means a single head must simultaneously model multiple attributes. In contrast, our approach is similar in spirit to a Mixture-of-Experts (MoE) design, where each attribute is modeled explicitly by a dedicated expert (i.e., a separate head). Consequently, the mixed-attribute behavior in the single-head setting leads to small but consistent performance drops across all metrics, demonstrating that the multi-head design better preserves attribute separation and improves robustness in prosody and acoustic modeling.

---

> > ### Author Response · Authors · 2025-11-28
> > **Official Comment by Authors**
> >
> > Dear Reviewer,
> >
> > We hope our responses have addressed your concerns. Please let us know if any additional clarification is needed. We truly value your feedback.
> >
> > Best regards,
> >
> > The Authors

---

### Official Review · Reviewer_MpaP · 2025-11-01

**Soundness:** 2
**Presentation:** 2
**Contribution:** 2
**Rating:** 2
**Confidence:** 3

**Summary:**

This paper proposes DiFlow-TTS, a zero-shot TTS system that performs purely discrete flow matching (DFM) over factorized speech tokens (content, prosody, acoustic). A Phoneme–Content Mapper (PCM) maps phonemes to discrete content tokens and embeddings (with duration/length regulation). A Factorized Discrete Flow Denoiser (FDFD) then predicts prosody and acoustic tokens via separate probability-velocity heads, conditioned on the content embeddings, a speaker embedding, and reference-prompt tokens. On LibriTTS training and LibriSpeech test-clean evaluation with 3-second prompts, the paper reports: UTMOS 3.98, WER 0.05, SIM-O 0.45 / SIM-R 0.54, F0 Acc 0.88 / RMSE 7.97, Energy Acc 0.73 / RMSE 0.007; and RTF 0.066 with 164 M params at NFE=16 (second-fastest behind a 1-step baseline). Ablations show large drops without speaker or content embeddings; NFE sweeps show strong gains up to ~8–16 steps with diminishing returns thereafter.

**Strengths:**

1. First reported discrete flow-matching approach to TTS with explicit factorization and separate prosody/acoustic heads.
 2. compact and fast relative to many contemporaries (RTF~0.066@16 NFE and ~164 M params); speed-quality trade-off is explored via NFE.

**Weaknesses:**

1. RTF comparisons are informative but would be stronger with fully standardized hardware conditions reported in-paper.
 2. The paper does not include controlled comparisons with strong discrete-token systems such as MaskGCT and IndexTTS2.
 3. The audio samples on the demo page sound relatively low in quality.

**Questions:**

1. I am familiar with several reported baselines. In Table 3, you list F5-TTS with WER = 0.24 (≈24%) at 32 NFEs, whereas the F5-TTS paper reports WER ≈ 2.42% (32 NFEs) and 2.53% (16 NFEs, RTF ≈ 0.15) on LibriSpeech-PC test-clean. These are an order of magnitude apart. Please precisely document your training/inference pipeline for each baseline (checkpoints used, NFE, solver, Sway Sampling on/off and schedule, CFG, ASR model/version, text normalization, prompt trimming/segmentation, random seeds, hardware), and provide a reproducible experiment sheet (per-model scripts/configs) so the community can replicate the exact numbers in your tables.
2. Factorized heads are reasonable, but the paper evaluates disentanglement indirectly. It lacks causal or information-theoretic evidence to substantiate controllability and orthogonality of factors.

---

> ### Author Response · Authors · 2025-11-28
> **Response to Reviewer MpaP [1/2]**
>
> **[W1]**: Clarification on hardware used to compute RTF
>
> RTF measurements were conducted on a server equipped with two AMD EPYC 7742 CPUs (128 cores, 256 threads) and 2.0 TiB RAM, along with an NVIDIA 80GB A100 GPU (details in Section C.2).
>
> ---
>
> **[W2]**: Controlled comparisons with strong discrete-token systems
>
> Thank you for your valuable feedback. We acknowledge that discrete-token systems such as MaskGCT and IndexTTS2 provide controllability options during inference. However, our goal in this work is not controllable TTS, which makes a direct comparison less appropriate. Our study focuses on investigating the viability of discrete flow matching for speech synthesis. As the first DFM-based speech synthesis model, DiFlow-TTS is designed for zero-shot TTS, where the input consists of a speech prompt whose prosody and speaker identity should be mimicked, along with the text to be synthesized.
>
> In this setting, the only controllable factor is the speech prompt itself. We already evaluate this aspect in Table 1, where our model demonstrates strong prosody transfer and outperforms all baselines, including MaskGCT, in prosody-related metrics. This indicates that DiFlow-TTS effectively captures and mimics the attributes present in the speech prompt, even without explicit controllability mechanisms. While MaskGCT and IndexTTS2 target a different problem formulation with user-specified controls, our results show that DiFlow-TTS is competitive and often superior within the zero-shot prompt-based TTS setting that we consider.
>
> ---
>
> **[W3]**: The audio samples on the demo page
>
> Thank you for your comment regarding the perceived quality of the audio samples. We acknowledge that subjective judgments of audio quality can vary depending on playback devices, listening environments, and individual perception. The MOS evaluation in Table 2 was designed to simulate the perceptual quality of the samples presented on the demo page, ensuring that the reported results capture an overall listener consensus rather than isolated impressions. Nevertheless, we sincerely appreciate the reviewer’s feedback and will further verify the audio rendering process to ensure consistent quality across playback settings.

---

> ### Author Response · Authors · 2025-11-28
> **Response to Reviewer MpaP [2/2]**
>
> **[Q1]**: Clarification on F5-TTS results
>
> We actually replicate the table in the OZSpeech [1], and we also adhere to the same LibriSpeech test set and evaluation protocol as benchmarked in that study. We have had discussions with OZSpeech authors to know the implementation of F5-TTS. Specifically, F5-TTS is reproduced strictly following the official F5-TTS repository instructions using the ``F5TTS_v1_Base`` configuration, the only different compared to original F5-TTS paper is amount of data, which is 500 hours of LibriTTS. During inference, it use  solver is Euler method, sway sampling coef is -1.0, CFG strength is 2.0. For the ASR system, it employed a CTC-based HuBERT ASR model pre-trained on LibriLight and fine-tuned on the 960-hour LibriSpeech training set. Text is normalized into lower case and remove punctuation, prompt trimming/segmentation, the LibriSpeech test-clean subset is filtered to retain utterances ranging from 4 to 10 seconds in duration, resulting in a total of 2.2 hours of audio. For each utterance, a prompt was randomly sampled from another utterance spoken by the same speaker, with the first 1, 3, or 5 seconds extracted to serve as the reference prompt.
>
> ---
> **[Q2]** : Clarification of disentanglement motivation
>
> We thank the reviewer for raising this important point. We clarify that our goal is to preserve the factorized structure provided by FaCodec within the generative framework, rather than to learn disentanglement from continuous representations. In our model, disentanglement is obtained directly from FaCodec, which explicitly separates prosody, content, and acoustic attributes.
>
> Our ablation study offers supportive evidence for this preservation. As shown in the last row of Table 5, replacing the multi-head prediction module with a single head forces the model to implicitly learn a joint distribution of prosody and acoustic attributes. This requires a single head to model multiple attributes simultaneously, whereas in our design each attribute is modeled explicitly by a dedicated expert head. The mixed-attribute behavior in the single-head configuration leads to small but consistent performance drops across all metrics, indicating that the multi-head design better maintains attribute separation and improves robustness in prosody and acoustic modeling.
>
> Furthermore, we conducted a finer-grained study by removing only the prosody attribute embedding or only the acoustic attribute embedding to provide deeper insights into how each attribute-specific signal contributes to the final synthesis quality. The results are shown in the table below. Removing either embedding produces clear degradations in naturalness and in metrics closely tied to the corresponding attribute. Attribute-type embeddings help the model identify which attribute each token represents when passed into the DiT. When an attribute embedding is removed, the model treats all tokens uniformly regardless of the multi-head structure, resulting in attribute entanglement and reduced synthesis quality.
>
> | Model                             | WER (%)  | UTMOS | SIM-O | F0 ACC | F0 RMSE | Energy ACC | Energy RMSE |
> |----------------------------------|------|-------|--------|--------|---------|-------------|--------------|
> | w/o Prosody Attribute Embedding  | 6.17 | 3.918 | 0.445  | 0.846  | 9.547   | 0.734       | 0.007        |
> | w/o Acoustic Attribute Embedding | 5.96 | 3.929 | 0.447  | 0.868  | 9.216   | 0.741       | 0.007        |
> | DiFlow-TTS                       | 4.83 | 3.978 | 0.454  | 0.884  | 7.972   | 0.735       | 0.007        |
>
> [1] Hieu et al. "OZSpeech: One-step Zero-shot Speech Synthesis with Learned-Prior-Conditioned Flow Matching" (ACL 2025)

---

> ### Author Response · Authors · 2025-11-28
> **Official Comment by Authors**
>
> Dear Reviewer,
>
> We hope our responses have addressed your concerns. Please let us know if any additional clarification is needed. We truly value your feedback.
>
> Best regards,
>
> The Authors

---

### Note · Authors · 2025-12-24

I have read and agree with the venue's withdrawal policy on behalf of myself and my co-authors.